# GEO-INVARIANT LEAD SCORING WITH DOMAIN-ADVERSARIAL TRANSFORMERS

## ABSTRACT

Predicting B2B lead conversion requires not only modeling long-range dependencies in richly sequenced customer interactions but also ensuring fair performance across under-represented geographies. While our DeepScore transformer backbone improved overall AUPR from 0.266 to 0.360, it exhibited significant geo-skew: majority-region (America) signals dominated feature learning (AUPR 0.474), leaving East-Asia (0.262) under-served. To address this, we embed a Domain-Adversarial Neural Network (DANN) module into DeepScore's architecture. A gradient-reversal layer connects a single geography discriminator to the shared transformer encoder, enforcing a minimax game that drives hidden representations to be predictive of conversion outcomes while remaining uninformative about geographic origin. Simultaneously, lightweight geo-specific classifier heads capture region-specific conversion patterns while maintaining shared feature representations, preventing model divergence across geographic markets. Geo-DANN DeepScore achieves a 4.3% relative gain in macro-AUPR and reduces inter-region AUPR gaps by up to 12.3%, all without degrading Americas accuracy. Empirically, we find that natural fixes such as geo-specific heads and inverse geo re-weighting can worsen performance in under-represented regions, whereas a simple Geo-DANN module improves East-Asia while maintaining or improving performance in the Americas.

## 1 INTRODUCTION

Predicting whether a prospective B2B lead will convert, known as *lead scoring*, is essential for allocating scarce sales resources effectively. For instance, enterprise cloud services sales typically require significant investment of Account Executive time for multiple lengthy meetings and technical demos with senior decision-makers, while Solution Architects need to develop custom proofs-of-concept and address specific technical requirements. Modern pipelines generate long sequences of time-stamped interactions (emails, ad clicks, webinars, calls), which tree-ensemble methods (XGBoost, LightGBM) struggle to model without extensive feature engineering. Transformer-based architectures, by applying self-attention over all touchpoints, have recently matched or surpassed these baselines on structured and sequential tasks (Lim et al., 2021b; Author, 2025), simplifying preprocessing while capturing long-range dependencies.

However, a single global model trained on pooled data often *privileges majority regions* (e.g., North America) and under-serves lower-volume markets (e.g., East-Asia), yielding significant geo-skew in performance metrics. To address this, we introduce **Geo-DANN DeepScore**: a transformer backbone augmented with (i) a gradient-reversal layer connecting a single geography discriminator that encourages representations to be invariant to geographic origin, and (ii) lightweight per-region classifier heads that learn residual local patterns on top of this shared representation. On a dataset of 1.4M leads across 10 geographic markets, Geo-DANN DeepScore achieves a 4.3% relative gain in macro-AUPR and reduces inter-region performance gaps by up to 12.3%, all without harming majority-region performance. This work is the first to apply adversarial domain adaptation at scale for B2B lead scoring, offering a practical blueprint for fair, high-fidelity predictions across heterogeneous markets.

In our deployment, overall conversion rates lie in the low single digits, and sales teams can realistically pursue only a fraction of incoming leads, making ranking quality at the top of the list critical. Because

a single global model is used across many regions, geographic disparities in model quality translate directly into disparities in sales opportunity: leads in under-served regions receive systematically worse scores, even when they represent equally promising prospects. We refer to this as a problem of *geo-fairness of utility*: all geographies should benefit comparably from the model's ranking quality, subject to maintaining high overall performance. Given the extreme class imbalance, we use area under the precision–recall curve (AUPR) as our primary threshold-free metric and conversion-rate Lift@30% to capture the business value of the top-ranked 30% of leads.

## 1.1 MOTIVATION AND PRIOR APPROACHES

Traditional lead scoring approaches have predominantly relied on region-specific model deployment strategies, wherein distinct predictive models are developed and maintained for individual geographic regions or business units. These conventional methods typically employ gradient boosting algorithms (such as LightGBM) and necessitate extensive feature engineering processes tailored to regional characteristics. While such region-specific modeling can effectively capture local market dynamics and behavioral patterns, this approach presents significant scalability and operational challenges. The maintenance of multiple parallel models substantially increases computational overhead, complicates feature pipeline architecture, and impedes systematic performance evaluation across regions. These operational inefficiencies and performance disparities highlight the need for a more unified and adaptable modeling framework.

## 1.2 THE DEEPSCORE ARCHITECTURE

To address these limitations, we developed **DeepScore**, a unified transformer-based architecture with three key design goals: (1) consolidate multiple regional models into a single, globally-deployable model, (2) leverage fine-grained sequential interaction data rather than aggregated tabular features, and (3) minimize reliance on manual feature engineering.

DeepScore processes customer interactions as sequences through a transformer encoder, analyzing both temporal patterns and textual content associated with each touchpoint. This approach captures the full customer journey, from initial website visits to email exchanges, without destroying sequential information through aggregation. Initial evaluations demonstrated substantial improvements over gradient boosting baselines (see Section 5.2).

However, analysis revealed significant regional performance disparities: high-volume regions with abundant training data saw improvements exceeding 60%, while lower-volume markets experienced gains below 20%. Such disparities stem from global loss minimization that naturally overfits to majority domains, a problem that standard class-imbalance techniques like SMOTE can actually exacerbate (Piccininni et al., 2024; van den Goorbergh et al., 2022; Carriero et al., 2025).

## 1.3 ALTERNATIVE APPROACHES AND THEIR LIMITATIONS

**Multi-Head Architecture.**   We initially explored a multi-head variant inspired by multi-task learning, where a shared transformer encoder feeds $G$ independent region-specific classifiers. The hypothesis was that each head would capture regional patterns while the shared backbone would benefit from pooled data. However, this approach yielded mixed results with a net decrease in macro-AUPR.

Three factors explain this failure: **(i) Gradient interference:** Back-propagation from high-volume regions dominates updates, causing the shared encoder to preferentially encode majority-region statistics (Yu et al., 2020; Standley et al., 2020). **(ii) Representation entanglement:** Without explicit regularization against geographic information, the encoder embeds regional cues that prevent effective cross-region transfer (Bousmalis et al., 2016; Ganin et al., 2016). **(iii) Sample inefficiency:** Region-specific heads receive limited mini-batches, leading to slow convergence and high gradient variance (Zamir et al., 2018; Ruder, 2017).

Given these limitations, we explored simpler rebalancing strategies before adopting the adversarial approach.

**Loss re-weighting strategies.** We also experimented with two re-weighting schemes: (i) inverse-frequency weighting based on regional training data availability ($w_g = N_{\text{total}}/N_g$), and (ii) inverse-performance weighting where regions are weighted proportionally to $1/\text{AUPR}_g$ from initial validation runs. As we show in Section 5.2, both variants reduced macro-AUPR and did not close gaps in low-volume regions. This confirms that changing the optimization emphasis alone is insufficient when the encoder still learns geography-specific patterns rather than geo-invariant features.

**Class rebalancing and other simple approaches.** Traditional rebalancing techniques (SMOTE, random over/undersampling), focal loss, progressive fine-tuning, and ensembles also failed to close geo gaps without unacceptable trade-offs in overall performance or operational complexity. These methods primarily treat geo imbalance as a sample-size problem and either discard large portions of data or overfit small regions, without addressing cross-geo distribution shift. Section 5.2 provides a detailed post-hoc analysis of why inverse weighting, in particular, harms East-Asia performance despite explicitly upweighting that region.

## 1.4 DOMAIN-ADVERSARIAL SOLUTION

The analysis above suggests that geographic performance gaps in DeepScore are driven primarily by *distribution shift and representation bias* rather than by simple class imbalance. Reweighting or resampling strategies (e.g., inverse-frequency and inverse-performance loss weighting, SMOTE-style oversampling) change which examples are emphasized during training but leave the underlying representation unchanged, and in our experiments they either degrade overall performance or worsen minority-region metrics. What we would like instead is a representation that preserves conversion-predictive signal while discarding geography-specific idiosyncrasies, so that data-rich regions can share useful structure with data-poor regions under a single global model.

To achieve this, we adopt domain-adversarial training (DANN) (Ganin et al., 2016) with geography as the domain. Geo-DANN DeepScore augments the DeepScore backbone with two components: (i) a set of geo-specific classifier heads $\{C_{\phi_g}\}_{g=1}^{G}$ on top of the shared representation $f_\theta(x)$, and (ii) a geography discriminator $D_\psi$ connected to $f_\theta(x)$ via a gradient-reversal layer. During training we minimize the lead-conversion loss while training $D_\psi$ to predict geography from $f_\theta(x)$ and training the encoder to *confuse* $D_\psi$. This min–max game encourages $f_\theta(x)$ to be predictive of conversion outcomes while being as uninformative as possible about geographic origin, so that the shared representation captures geo-invariant patterns and the geo-specific heads learn modest residual adjustments. Section 3 details this architecture, including the GRL schedule and discriminator parameterization.

Geo-DANN is attractive in our deployment setting for three reasons. First, it directly targets the root cause of the observed imbalance—cross-geo distribution shift—rather than class imbalance, and thus complements rather than competes with the class-imbalance mitigation techniques already used in our pipeline. Second, it preserves the operational simplicity of a single globally-deployable model while enabling cross-geo knowledge transfer from data-rich to data-poor markets. Third, as we show in Section 5.2, naively increasing model capacity improves performance in the data-rich Americas region but widens geographic gaps, whereas with Geo-DANN the same capacity increases improve both Americas and low-volume regions and reduce inter-geo gaps. Section 4 interprets this behavior through a standard domain-adaptation bound.

**Contributions.** Domain-adversarial training (DANN) is a standard tool for domain adaptation (Ganin et al., 2016), and domain-adaptation theory provides conditions under which learning domain-invariant representations improves target-domain error (Ben-David et al., 2010). Our contribution is using domain-adversarial training to mitigate *geographic performance disparities* in a large-scale *sequential lead-scoring model*, enabling a single model to learn trends from well represented geos in a way that benefits those that are poorly represented. Concretely, we:

(1) **Diagnose why standard fixes fail to improve under-represented regions in a real global lead-scoring system.** We show that a strong transformer-based lead-scoring model, trained as a single global model, exhibits substantial differences in AUPR and Lift@30% across geographies: data-rich regions such as Americas benefit the most, while low-volume regions lag behind. We systematically evaluate natural fixes—multi-head modeling, loss re-weighting (inverse-frequency and inverse-performance), and standard class rebalancing techniques (e.g., SMOTE)—and find that

they either degrade overall performance or *worsen* minority-region metrics. This analysis supports a key insight of our work: geographic performance gaps in this setting stem primarily from distribution shift and representation bias, not mere class imbalance.

(2) **Propose Geo-DANN DeepScore for cross-geo knowledge transfer and geo-fairness.** We introduce Geo-DANN DeepScore, a simple architecture that combines a shared interaction-sequence encoder, lightweight geo-specific classifier heads, and a single geography discriminator trained via a gradient-reversal layer. This design encourages the shared representation to be geo-invariant while allowing modest per-region adjustments, enabling signal to transfer from data-rich to data-poor markets and supporting a single globally-deployable model with reduced operational complexity compared to region-specific baselines.

(3) **Provide a deployment-scale empirical evaluation and ablations.** On 1.4M leads across 10 geographies, Geo-DANN DeepScore improves macro-AUPR by 4.3% and reduces inter-geo AUPR gaps by up to 12.3% without degrading Americas performance. We compare against a production LightGBM model, single-head and multi-head DeepScore variants, and loss re-weighting baselines, and study the behavior of Geo-DANN under model-capacity scaling and adversarial design choices (GRL schedule, conditional discriminator). Across these experiments, the proposed configuration consistently improves both majority and minority regions and narrows geographic gaps. We further show that, unlike the baseline multi-head model, Geo-DANN allows us to increase encoder capacity while improving performance in both Americas and low-volume regions, instead of widening geographic performance gaps.

(4) **Interpret Geo-DANN through a domain-adaptation lens.** Using a standard domain-adaptation bound (Ben-David et al., 2010) as a guiding abstraction, we view Geo-DANN as encouraging the shared encoder to reduce cross-geo divergence while geo-specific heads capture irreducible regional differences. This provides a conceptual explanation for why the same architecture can both raise overall performance and narrow geographic gaps, without changing the class-imbalance structure of the data.

## 2 RELATED WORK

### 2.1 THE EVOLUTION OF LEAD SCORING

Lead scoring has evolved from rule-based systems encoding sales heuristics (D'Haen et al., 2013) to modern machine learning approaches. The current industry standard commonly employs gradient boosting methods, XGBoost (Chen & Guestrin, 2016) and LightGBM (Ke et al., 2017), due to their strong performance on tabular data. These methods typically require extensive feature engineering and aggregation, which can destroy the sequential nature of customer journeys (Khurana et al., 2018).

Recent work has explored deep learning for lead scoring, but primarily through simple neural networks on the same aggregated features (González-Flores et al., 2025). The critical insight, that the journey itself is the signal, has been overlooked in favor of incrementally improving feature engineering.

### 2.2 GEO-AWARE AND CROSS-DOMAIN MODELING

Geo-aware modeling in traffic forecasting, environmental prediction, and large-scale recommendation often uses a shared encoder with lightweight local heads or city/domain-specific adaptations (Yu et al., 2018; Zhang et al., 2025; Ma et al., 2024; Xu et al., 2022; Wang et al., 2022). These approaches show that shared representations plus modest local specialization can capture spatial heterogeneity without training separate models for each region. Our work applies a similar high-level idea in a B2B lead-scoring setting, but combines it with explicit domain-adversarial alignment for geography rather than relying solely on multi-head architectures.

### 2.3 DOMAIN-ADVERSARIAL LEARNING

Ganin et al. (2016b) introduced Domain-Adversarial Neural Networks (DANN), embedding a gradient-reversal layer and domain discriminator into back-propagation so that feature extractors become domain-confusing while label predictors remain discriminative. Ben-David et al. (2010)

provided generalization bounds for domain adaptation in terms of an $\mathcal{H}\Delta\mathcal{H}$ divergence between source and target distributions. Subsequent work has extended DANN to high-capacity backbones in vision and remote sensing, and highlighted the importance of allowing some domain-specific modeling when conditional distributions differ (Zhao et al., 2019). Related fairness-aware training approaches such as Group DRO and IRM optimize worst-group performance or invariance penalties across groups, and Wasserstein variants such as WDGRL modify the divergence term; adapting these methods to our proprietary, large-scale sequential setting with many overlapping groups and strict training budgets is an important direction for future work. We adopt the standard DANN formulation with geography as the domain and geo-specific heads on top of a shared encoder; technical details and our domain-adaptation perspective are presented in Sections 3 and 4.

Transformers have also been widely adopted for heterogeneous sequential and structured/tabular data, including Temporal Fusion Transformers and TabTransformer, as well as numerous variants for classification and tabular representation learning (Lim et al., 2021a; Huang et al., 2020; Devlin et al., 2019; Liu et al., 2019; Badaro et al., 2023; Somvanshi et al., 2024; Ruan et al., 2024; Gorishniy et al., 2021; Arík & Pfister, 2021; Singh et al., 2023; Wang & Sun, 2022; Chen et al., 2024; Fan & Waldmann, 2024). DeepScore follows this line by applying self-attention over long customer-journey sequences with static profile features; Appendix A.2 provides additional background on this literature.

**Geographic fairness and B2B lead scoring.** Models trained on data from a few dominant regions often exhibit substantial performance drops in under-represented geographies, raising fairness concerns in vision, language, and recommendation systems. Much of this work focuses on robustness or average target performance. In contrast, we focus on *fairness of utility* across geographies in a business decision system: the goal is to reduce performance disparities between high-volume and low-volume regions under a single global model, rather than to optimize any one region in isolation. Prior work on B2B lead and opportunity scoring has largely focused on overall predictive accuracy or uplift and does not explicitly address cross-geo domain shift or fairness.

## 3 METHODOLOGY

### 3.1 DEEPSCORE BACKBONE

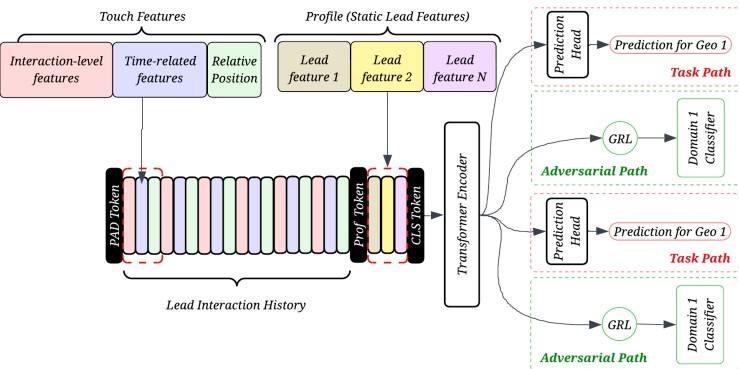

Figure 1: DeepScore Architecture (main)

DeepScore represents each lead as a sequence of timestamped marketing and sales interactions (emails, ads, calls, webinars) together with a static profile. For each touch we learn embeddings for metadata (channel, product, etc.), text, and relative time; these are concatenated into a single "touch" vector. A profile embedding summarizes static categorical and numerical attributes of the account.

We prepend the profile token and feed the padded sequence into a transformer encoder, using standard positional and time embeddings. The encoder's [CLS]-style token serves as the dense lead representation $(f_\theta(x))$, which is mapped by a linear layer to the conversion logit. Full architectural details, including feature lists, embedding sizes, and layer dimensions, are provided in Appendix E.

## 3.2 MULTI-HEAD PREDICTION DESIGN

To improve the models ability to focus on distinct geos, we replace the single linear classifier by a `torch.nn.ModuleDict` that stores ten single-layer MLP heads. During a forward pass we look up the geography ID for every lead, select the corresponding head, and compute its logits, where $C$ is the individualized classification heads:

$$\texttt{logits}_i = C_{\phi_{g(i)}}\big(f_\theta(x_i)\big), \quad g(i) \in \{1, \dots, 10\}.$$

## 3.3 GEO-DANN - DOMAIN–ADVERSARIAL NEURAL NETWORK

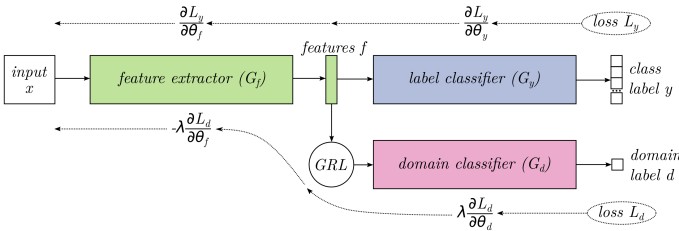

Figure 2: Information flow with a Gradient-Reversal Layer (GRL). During back-propagation the domain-loss gradient is sign-flipped, forcing the encoder to learn geography-invariant features while the task-loss gradient is propagated unchanged, following the formulation of Ganin et al. (2016a). Diagram from ResearchGate (2025).

The Geo-DANN module injects an adversarial game into the otherwise supervised training of DeepScore. Its goal is to learn features that remain predictive of conversion while being *maximally confusing* with respect to geography. We implement three key components:

**(1) Adversarial principle:** Our approach follows the min–max framework of Ganin et al. (2016)—we minimize the lead-conversion binary cross-entropy loss while maximizing the error of a geography discriminator $D_\psi$ that attempts to recover the geography label from the shared representation $f_\theta(x)$. This adversarial game encourages the encoder to remove geography-specific information from $f_\theta(x)$ while preserving conversion-predictive signal. Combined with geo-specific classifier heads on top of $f_\theta(x)$, this yields a representation that is approximately invariant across geographies while still allowing modest regional specialization.

Formally, let $\mathcal{X}$ denote the input space and $\mathcal{Y} = \{0, 1\}$ the binary conversion label. Each training example $(x_i, y_i, g_i)$ consists of features $x_i \in \mathcal{X}$, a conversion label $y_i \in \mathcal{Y}$, and a geography $g_i \in \{1, \dots, G\}$. The encoder $f_\theta$ maps $x_i$ to a shared representation $\mathbf{h}_i = f_\theta(x_i) \in \mathbb{R}^d$. We attach a geo-specific classifier $C_{\phi_g}$ for each geography $g$, which produces a logit $z_i = C_{\phi_{g_i}}(\mathbf{h}_i)$ and a conversion probability $\hat{y}_i = \sigma(z_i)$. The task loss is the binary cross-entropy

$$\mathcal{L}_{\text{task}} = -\frac{1}{N} \sum_{i=1}^{N} \big[ y_i \log \hat{y}_i + (1 - y_i) \log(1 - \hat{y}_i) \big]. \tag{1}$$

The geography discriminator $D_\psi$ takes $\mathbf{h}_i$ as input and outputs a softmax distribution $p_\psi(g \mid \mathbf{h}_i)$ over the $G$ regions. The domain loss is

$$\mathcal{L}_{\text{domain}} = -\frac{1}{N} \sum_{i=1}^{N} \log p_\psi(g_i \mid \mathbf{h}_i). \tag{2}$$

We optimize a combined objective

$$\mathcal{L}_{\text{total}}(t) = \mathcal{L}_{\text{task}} + \alpha(t) \, \mathcal{L}_{\text{domain}}, \tag{3}$$

where $\alpha(t) \in [0, 1]$ is a time-dependent weight on the domain loss. Because a gradient-reversal layer multiplies the gradient of $\mathcal{L}_{\text{domain}}$ by $-\lambda(t)$ when it flows into the encoder, the effective encoder update at step $t$ is

$$\nabla_\theta \mathcal{L}_{\text{encoder}} = \nabla_\theta \mathcal{L}_{\text{task}} - \alpha(t) \lambda(t) \, \nabla_\theta \mathcal{L}_{\text{domain}}. \tag{4}$$

Thus the encoder is encouraged to both reduce conversion error and confuse the geography discriminator, learning representations that are predictive of conversion but less informative about geography.

**(2) Gradient-reversal layer (GRL):** Instead of alternating optimization, the adversarial element is optimized simultaneously in a *single* back-prop pass by inserting a **gradient-reversal layer** (`GradRev`) between the encoder and $D_\psi$. First proposed by Ganin & Lempitsky (2015) and later popularized in computer vision (Tzeng et al., 2017; Shen et al., 2018), GRL is the identity in the forward pass but multiplies incoming gradients by $-\lambda_{\mathrm{GRL}}$ on the backward pass. This simple trick lets us leverage off-the-shelf optimizers and preserves the speed of standard training loops. The detailed implementation with exact layer specifications is shown in Figures 4 and 7.

**Adversarial schedules.** We follow the standard curriculum of Ganin et al. (2016) and use a time-dependent weight on the domain loss: a logistic annealing schedule for the gradient-reversal strength combined with a short warm-up period early in training. This prevents the adversary from overpowering the encoder at the start while gradually encouraging geography-invariant representations. The exact formulas and hyperparameters for $\lambda(t)$ and $\alpha(t)$ are given in Appendix B.1.

**Discriminator parameterization and CDAN ablation.** The geography discriminator $(D_\psi)$ is a small two-layer MLP with dropout. Appendix B.2 provides architectural details and describes a conditional domain-adversarial (CDAN) variant that we evaluate as an ablation; its performance is similar to Geo-DANN on our data.

## 4 DOMAIN-ADAPTATION PERSPECTIVE: WHY DANN LEVELS GEO PERFORMANCE

Let $P_S$ and $P_T$ denote the source and target distributions over $(X, Y)$ for two geographies, with a common hypothesis class $\mathcal{H}$. Ben-David et al. (Ben-David et al., 2010) show that the target error $\epsilon_T$ of a hypothesis $h \in \mathcal{H}$ can be bounded as

$$\epsilon_T(h) \ \leq \ \epsilon_S(h) \ + \ \tfrac{1}{2}d_{\mathcal{H}\Delta\mathcal{H}}(P_S, P_T) \ + \ \lambda^*, \tag{5}$$

where $\epsilon_S(h)$ is the source error, $d_{\mathcal{H}\Delta\mathcal{H}}$ measures how easily hypotheses in $\mathcal{H}$ can distinguish $P_S$ from $P_T$, and $\lambda^*$ is the error of the optimal joint hypothesis on both domains.

In our setting, we treat the data-rich Americas region as the source domain and low-volume regions (e.g., Europe and East-Asia) as target domains. The geography discriminator $D_\psi$ in Geo-DANN approximates the $d_{\mathcal{H}\Delta\mathcal{H}}$-divergence between these domains, and the gradient-reversal training objective encourages the encoder to reduce this divergence, making $f_\theta(x)$ increasingly geo-invariant. From this perspective, simply increasing model capacity on the source domain (Americas) tends to reduce the source error term but can increase the $d_{\mathcal{H}\Delta\mathcal{H}}$ term, widening geographic gaps. By adding the DANN objective, we explicitly penalize this divergence, so increasing capacity can simultaneously reduce source error and cross-geo divergence in our setting, matching the empirical behavior observed in Section 5.2.2. Attaching small geo-specific classifier heads on top of the shared representation then allows modest residual regional adjustments without enforcing full invariance; Appendix I provides a more detailed discussion.

## 5 RESULTS

### 5.1 DATA

To compare with the production LIGHTGBM baseline, we train all models on the same two-year window of marketing-qualified leads (May 2022–May 2024) from ten regional business units, comprising 1.4M labeled examples, and evaluate on a subsequent 2.5-month hold-out period. Features are computed only from events observed before each lead's qualification time so that no post-decision information leaks into training or evaluation.

We report average precision–recall (AUPR) and Conversion-Rate Lift@30%, which quantify overall ranking quality and relative conversion rate within the top-ranked 30% of leads. We use macro-AUPR together with per-geo AUPR and Lift@30% to study both aggregate performance and geographic disparities; further details of the temporal split and base-rate prevalence are provided in Appendix H.

## 5.2 Quantitative analysis

Our primary goal is not only to maximize overall ranking quality, but to *reduce geographic performance disparities* under a single global model. We therefore evaluate each model using both: (i) macro-AUPR across regions and (ii) simple geo-fairness metrics derived from per-geo AUPR and Lift@30%. In particular, we consider the inter-geo AUPR gap $\text{Gap}_{\text{AUPR}} = \max_g \text{AUPR}_g - \min_g \text{AUPR}_g$ and relative minority ratios such as $\text{AUPR}_{\text{East-Asia}}/\text{AUPR}_{\text{Americas}}$, which are implicitly summarized in Tables 5.2 and 5.2. A model is more geo-fair in our sense if it achieves high macro-AUPR while reducing $\text{Gap}_{\text{AUPR}}$ and improving minority-region ratios.

**Baselines.** We compare Geo-DANN DeepScore against the following baselines: (i) a feature-engineered LightGBM model representative of current practice; (ii) **Single-Head DeepScore**, which applies the transformer encoder and a single global classifier head to all geographies; (iii) **Multi-Head DeepScore**, which shares the encoder but uses one classifier head per geography without any adversarial loss; and (iv) two reweighting variants of Single-Head DeepScore based on inverse geography frequency and inverse per-geo performance. All models share the same encoder architecture and training budget; differences arise only from the heads and loss terms. Variants of the adversarial objective (including a CDAN-style conditional loss) are treated as ablations and reported in Appendix D.

Table 1: Average Precision-Recall (AUPR) performance across geographic regions

| Model | Macro | East-Asia | Europe | Americas |
|---|---|---|---|---|
| DeepScore DANN | **0.360** | **0.288** | **0.271** | **0.474** |
| DeepScore Multi-Head | 0.345 | 0.262 | 0.258 | 0.459 |
| DeepScore Single-Head | 0.350 | 0.270 | 0.255 | 0.464 |
| Benchmark (LightGBM) | 0.266 | 0.249 | 0.227 | 0.287 |
| Inverse frequency weighting | 0.356 | 0.179 | 0.247 | 0.448 |
| Inverse performance weighting | 0.343 | 0.177 | 0.265 | 0.469 |

Table 2: Relative AUPR improvements across regions compared to Americas

| Model | East-Asia | Europe |
|---|---|---|
| DANN | **0.625** | **0.572** |
| Multi-Head | 0.572 | 0.562 |
| Single-Head | 0.582 | 0.549 |

Table 3: Conversion rate Lift@30% across regions

| Model | Macro | EU | E-Asia | AM |
|---|---|---|---|---|
| DANN | **2.510** | **2.589** | 2.294 | **2.485** |
| M-Head | 2.470 | 2.535 | 2.401 | 2.458 |
| S-Head | 2.465 | 2.501 | **2.416** | 2.450 |

Table 5.2 shows that adding the domain-adversarial objective (+ DANN) lifts *macro* AUPR from $0.345$ to $0.360$, an absolute gain of $+0.015$ or $+4.3\%$ over the strongest non-adversarial baseline (multi-head). The improvement is driven largely by closing the gap in under-represented regions: East-Asia increases from $0.221$ to $0.235$ $(+6.3\%)$ and Europe from $0.258$ to $0.271$ $(+5.0\%)$. Performance in the data-rich Americas market is *increased* from $0.459$ to $0.474$ $(+3.2\%)$, indicating that the adversarial pressure does not harm the majority domain.

A similar pattern appears in the business-facing *Conversion-Rate Lift30* (Table 3, right). The DANN raises the macro lift from $2.47$ to $2.51$ and yields the highest lift in *every* geography. Because sales teams operate under quota constraints, even a $1\%$–$2\%$ relative lift in the top-ranked segment translates into a measurable increase in bookings.

**Why America also improves.** One might expect adversarial alignment to trade off accuracy in the majority domain for gains elsewhere, yet America AUPR rises (Table 5.2). Two factors explain this behavior.

First, **regularization via noise injection:** The gradient-reversal signal imposes an additional constraint on the encoder—features that overfit America-specific artifacts (e.g. US holiday spikes, region-specific email templates) are actively penalized. This behaves like a structured noise injection,

discouraging brittle correlations and acting as a form of regularization. Empirically we observe a $3.5\%$ reduction in the generalization gap between training and validation loss for America, suggesting that the adversary mitigates mild overfitting and therefore *improves* true America performance.

Second, **specialized head retains local signal:** Although the shared representation is geography-agnostic, the America-specific prediction head is free to relearn legitimate local patterns. In practice it captures macro-economic cycles and channel saturation effects unique to the American funnel, while benefiting from the cleaner, less noisy feature space delivered by the adversary. The combination of a *robust* encoder plus a *flexible* local head yields the modest yet consistent lift observed in every offline fold. Prior work shows that predictive lead scoring improves sales performance and ROI, and that slow response to leads leads to large conversion losses (Wu et al., 2023; Oldroyd et al., 2011).

**Why Inverse weighting is not helping**   Our experiments reveal a significant degradation in East-Asia performance under inverse weighting approaches compared to both DeepScore variants. Specifically, the inverse frequency weighting model achieves an AUPR of only 0.179 in East-Asia, substantially lower than DeepScore Single-Head (0.270) and Multi-Head (0.262). This counter-intuitive result, where explicit compensation for data imbalance actually harms minority region performance, can be attributed to three key factors: First, aggressive upweighting of sparse East-Asia samples amplifies noise and region-specific outliers, leading to unstable gradient updates during training. Second, the weighting mechanism fails to address the fundamental distribution shift between regions, merely adjusting sample importance without learning truly transferable features. Third, and most critically, our East-Asia training data likely suffers from selection bias or data collection inconsistencies that create a distribution shift between training and test sets. This hypothesis is confirmed by examining train-validation versus test performance: East-Asia shows a train-val AUPR of 0.290 but drops to 0.179 on test data under inverse weighting, while Americas maintains consistent performance (train-val: 0.412, test: 0.448). The unweighted model shows much smaller train-test gaps across all regions.

When unweighted, the model largely ignores these corrupted samples and successfully transfers patterns learned from cleaner Americas/Europe data. However, inverse weighting forces the model to memorize these non-representative East-Asia training patterns, causing catastrophic failure on the properly-distributed test set. This explains why the unweighted model (0.270) significantly outperforms the weighted version (0.179). This finding reinforces our theoretical analysis that geographic performance gaps stem from distribution misalignment rather than simple class imbalance, highlighting why our DANN-based approach, which explicitly optimizes for domain-invariant representations, proves more effective. By learning geography-invariant features, DANN sidesteps potentially corrupted regional signals entirely, explaining its superior performance (0.288) even compared to the unweighted baseline.

**Calibration.**   We also assess probability calibration using Brier scores (Appendix D.4). Geo-DANN maintains calibration comparable to the non-adversarial Multi-Head DeepScore model across Americas, Europe, and East-Asia while substantially reducing inter-geo performance gaps in AUPR and Lift@30%.

### 5.2.1   ABLATION: GRL SCHEDULE

We ablate the GRL schedule parameter $\gamma$. Table 5 (Appendix) reports macro-AUPR and regional AUPR for $\gamma \in \{2, 5, 10\}$. Performance is fairly stable across this range; lower values ($\gamma = 2$) show minimal differences from the baseline. For the main results we therefore use $\gamma = 10$ as a simple and robust choice.

### 5.2.2   MODEL CAPACITY, GEOGRAPHIC DISPARITY, AND THE EFFECT OF GEO-DANN

We next study how model capacity affects geographic performance with and without Geo-DANN. Figure 3 shows AUPR performance across regions as a function of model complexity (measured in millions of parameters) for three architectures: Single-Head DeepScore, Multi-Head DeepScore, and Multi-Head + DANN (Geo-DANN DeepScore).

Without DANN (left and center panels), increasing model capacity maintains substantial performance gaps between regions: Americas (green) consistently outperforms East-Asia (red) and Europe (purple)

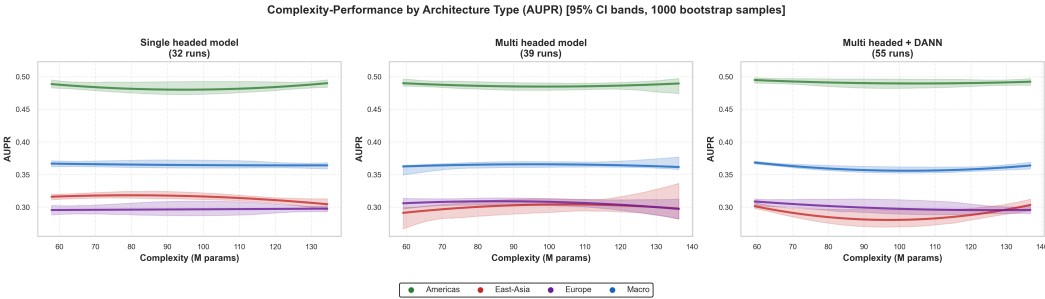

Figure 3: Complexity-Performance by Architecture Type. AUPR performance across geographic regions as a function of model complexity (parameters in millions) for three architectures: Single-Head DeepScore (left), Multi-Head DeepScore (center), and Multi-Head + DANN (right). Shaded bands show 95% confidence intervals from 1000 bootstrap samples. Without DANN, increasing capacity maintains or widens inter-region gaps. With DANN, the gap between Americas and East-Asia/Europe narrows substantially while maintaining strong overall performance.

by significant margins, and these gaps persist or even widen with increased capacity. The single-head model shows the most stable but widely separated performance curves, while the multi-head model exhibits some convergence at lower capacities but maintains clear regional disparities.

With Geo-DANN (right panel), the regional performance curves converge dramatically. The gap between Americas and East-Asia/Europe narrows from approximately 0.20 AUPR points to less than 0.05, while macro performance (blue) remains competitive or improves. This empirical behavior mirrors the domain-adaptation perspective in Section 4: unconstrained capacity reduces source error but increases divergence between source and target geographies, whereas the adversarial objective explicitly penalizes this divergence, allowing larger models to improve performance across regions in our setting rather than further amplifying disparities.

## 6 CONCLUSION

DeepScore with the geo-DANN module combines transformer sequence modeling with domain-adversarial alignment to deliver state-of-the-art lead-conversion prediction across heterogeneous geographies while narrowing performance gaps without harming majority domains, laying groundwork for applying adversarial adaptation to other business-critical models such as churn or lifetime-value estimation. Our results indicate that commonly recommended interventions such as inverse geo re-weighting can severely hurt low-volume regions under distribution shift, while domain-adversarial alignment makes scaling a single global model much safer across geographies. For comprehensive surveys on domain adaptation theory and deep unsupervised domain adaptation methods, see Redko et al. (2020) and Wilson & Cook (2020).

**Limitations and future work.** Our study has several limitations. First, the data are proprietary and drawn from a single organization; while we provide detailed architectural and training descriptions, we cannot release the dataset, per-geo prevalence, or full confusion matrices for confidentiality reasons. Second, we focus on geography as the primary domain attribute and study only one family of adaptation methods (DANN and a CDAN variant) on a specific transformer backbone; we do not implement group-robust or invariant training objectives such as Group DRO, IRM, or WDGRL, and leave a systematic comparison with these approaches to future work. Third, we evaluate temporal robustness on a single hold-out period aligned with the median qualification-to-opportunity lag; broader evaluation across multiple future periods is an important direction. Finally, we evaluate geo-fairness primarily through ranking metrics (AUPR and Lift@30%) and Brier scores; a more detailed calibration and threshold-based error analysis across additional axes is left for future work.

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

## A  Additional Related Work

### A.1  Geo-aware and Cross-Domain Modeling

Global platforms and scientific applications have adopted a variety of strategies to balance shared learning with local adaptation. In traffic forecasting, Spatio-Temporal Graph Convolutional Networks (STGCN) learn a single graph-based encoder over an entire road network, while more recent evolutionary GNNs dynamically update region-specific adjacency structures to capture local traffic patterns without training separate models for each city (Yu et al., 2018; Ma et al., 2024). In large-scale recommendation systems, two-tower embedding architectures train a global user-item model that is then fine-tuned or re-indexed at the city level, reducing the need for thousands of city-specific models and lowering operational overhead (Wang et al., 2022)

Environmental and spatial forecasting have employed multi-task frameworks with a shared backbone plus lightweight domain-specific heads, demonstrating improved spatial generalization while avoiding fully separate regional models (Zhang et al., 2025). CDTrans combines self-attention pseudo labels to close domain gaps across diverse image datasets, offering a blueprint for single-model multi-domain deployment without per-domain retraining (Xu et al., 2022).

### A.2  Transformers for Sequential and Structured Data

Extending transformers beyond text requires handling heterogeneous inputs and irregular timing. Temporal Fusion Transformers (Lim et al., 2021a) introduced specialized components for time series forecasting with static covariates. TabTransformer (Huang et al., 2020) applies attention to tabular features but doesn't model true sequences. Most relevant is the line of work on multi-modal transformers (Tsai et al., 2019), though these typically handle aligned modalities (e.g., video and audio) rather than the diverse interaction types in customer journeys.

Since their introduction for sequence modeling, transformers have been broadly adopted for classification across multiple modalities. In natural language processing, pretrained encoders such as BERT fine-tuned on GLUE benchmarks have set new state-of-the-art on sentence and document classification tasks Devlin et al. (2019); Liu et al. (2019).

TabTransformer first demonstrated that self-attention contextualizes categorical embeddings to rival tree-ensemble models on tabular tasks Huang et al. (2020). Subsequent surveys chronicle dozens of transformer-based variants for tabular representation learning Badaro et al. (2023); Somvanshi et al. (2024); Ruan et al. (2024); Gorishniy et al. (2021); Arík & Pfister (2021); Singh et al. (2023); Wang & Sun (2022); Chen et al. (2024); Fan & Waldmann (2024), but none address domain shift in business-scale B2B marketing sequences at the scale tackled here.

## B  Geo-DANN Implementation Details

### B.1  Adversarial Schedules

If the adversary is too strong too early, the encoder may collapse to features that are useless for conversion; if too weak, it never removes geographic artifacts. We therefore follow the logistic curriculum of Ganin et al. (2016) and apply both a schedule on the gradient-reversal strength and a warmup on the domain loss.

Let $t$ denote the current training step and $T$ the total number of steps, and define normalized training progress $p(t) = \min(1, t/T)$. We set

$$\lambda(t) = \frac{2}{1 + \exp(-\gamma\, p(t))} - 1, \tag{6}$$

with steepness parameter $\gamma = 10$. Early in training $\lambda(t) \approx 0$, allowing the encoder to discover predictive structure; as $p(t) \to 1$ the term approaches 1, steadily raising the adversarial pressure.

In addition, we apply a linear warmup to the domain-loss weight:

$$\alpha(t) = \min\left(1, \frac{t}{W}\right), \tag{7}$$

where $W = E_w \times B$ is the number of warmup steps, $E_w$ is the number of warmup epochs (we use $E_w = 3$), and $B$ is the number of batches per epoch. If $W = 0$ then $\alpha(t) = 1$ for all $t$. The product $\alpha(t)\lambda(t)$ therefore controls the total adversarial pressure on the encoder over training.

### B.2 Discriminator and CDAN Variant

The geography discriminator $D_\psi$ is a two-layer multilayer perceptron with ReLU activations and dropout rate 0.2 between hidden layers. In some runs we apply spectral normalization to the discriminator weights, $\bar{W} = W/\sigma(W)$ where $\sigma(W)$ is the largest singular value, to prevent the discriminator from overpowering the encoder during adversarial training; this improves stability but has little effect on final performance.

We also evaluated a conditional domain-adversarial variant (CDAN; Long et al. 2018) as an ablation. In this case the discriminator is conditioned on both the shared representation and the task predictions via the outer product $\mathbf{g}_i = \mathrm{softmax}(C_{\phi_{g_i}}(\mathbf{h}_i)) \otimes \mathbf{h}_i$, enabling class-conditional domain alignment. As discussed in Section D, CDAN achieves similar macro-AUPR and inter-geo AUPR gaps to Geo-DANN on our data.

## C Implementation details for Geo-DANN

For completeness we record the exact schedules and hyperparameters used in Geo-DANN DeepScore. The gradient-reversal strength $\lambda(t)$ follows the logistic schedule

$$\lambda(t) = \frac{2}{1 + \exp(-\gamma\, p(t))} - 1, \quad p(t) = \min(1, t/T), \tag{8}$$

with steepness parameter $\gamma = 10$ and $T$ the total number of training steps. The domain-loss weight uses a linear warmup

$$\alpha(t) = \min\left(1, \frac{t}{W}\right), \tag{9}$$

where $W = E_w X B$ is the number of warmup steps, $E_w$ is the number of warmup epochs, and $B$ is the number of batches per epoch. If $W = 0$ then $\alpha(t) = 1$ for all $t$. The effective adversarial gradient to the encoder is

$$\nabla_\theta \mathcal{L}_{\mathrm{encoder}} = \nabla_\theta \mathcal{L}_{\mathrm{task}} - \alpha(t)\, \lambda(t)\, \nabla_\theta \mathcal{L}_{\mathrm{domain}}. \tag{10}$$

The geography discriminator $D_\psi$ uses two hidden layers with ReLU activations and 0.2 dropout; when enabled, spectral normalization rescales each weight matrix as $\bar{W} = W/\sigma(W)$, where $\sigma(W)$ is the largest singular value, which stabilizes adversarial training in our setting.

## D CDAN ablation

In preliminary experiments we evaluated a conditional domain-adversarial objective (CDAN; (Long et al., 2018)), where the discriminator receives both the shared representation $f_\theta(x)$ and the classifier logits as input, allowing it to model the interaction between predicted labels and domain features.

Table 4 compares Geo-DANN DeepScore with a CDAN-style conditional adversarial objective on the same encoder and geo-specific heads. Across all regions, Geo-CDAN achieves slightly lower macro-AUPR (0.325 vs. 0.331) with higher variance, while maintaining comparable inter-geo AUPR gaps. The East-Asia region shows a marginal improvement with CDAN (0.302 vs. 0.299) but this is within the standard error. CDAN does not change the qualitative capacity–fairness behavior, and for simplicity and stability, we adopt the standard DANN formulation as the default adversarial objective in Geo-DANN DeepScore.

### D.1 GRL schedule ablation

Table 5 reports the impact of different GRL schedules on macro-AUPR and regional AUPR. Performance remains relatively stable across the tested range, with $\gamma = 10$ and $\lambda_{\mathrm{max}} = 1.0$ providing a good balance between convergence and stability.

Table 4: Ablation of adversarial objective: DANN vs. CDAN. Macro AUPR and AUPR for Americas and East-Asia on the held-out test set.

| Model | Macro AUPR | AUPR (Americas) | AUPR (East-Asia) |
|---|---|---|---|
| Geo-DANN | $0.331 \pm 0.010$ | $0.489 \pm 0.008$ | $0.299 \pm 0.019$ |
| Geo-CDAN (ablation) | $0.325 \pm 0.015$ | $0.486 \pm 0.010$ | $0.302 \pm 0.007$ |

Table 5: GRL schedule ablation. Macro AUPR and Americas AUPR and East-Asia AUPR and Europe AUPR on the held-out test set.

| Schedule ($\gamma, \lambda_{\max}$) | Macro AUPR | Americas AUPR | East-Asia AUPR | Europe AUPR |
|---|---|---|---|---|
| (10, 1.0) | $0.308 \pm 0.007$ | $0.476 \pm 0.006$ | $0.298 \pm 0.002$ | $0.266 \pm 0.012$ |
| (2, 1.0) | $0.309 \pm 0.008$ | $0.478 \pm 0.006$ | $0.298 \pm 0.001$ | $0.266 \pm 0.012$ |
| (5, 1.0) | $0.310 \pm 0.008$ | $0.479 \pm 0.006$ | $0.296 \pm 0.005$ | $0.270 \pm 0.010$ |

### D.2 COMPREHENSIVE DANN PARAMETER ABLATION

Table 6 presents a comprehensive ablation study of DANN parameters, demonstrating the model's sensitivity to discriminator architecture and training dynamics.

Table 6: Comprehensive DANN parameter ablation study.[*]

| Parameter | Setting | Macro AUPR | Americas AUPR | East-Asia AUPR | Europe AUPR |
|---|---|---|---|---|---|
| Disc Dims | [128, 128] | $0.315 \pm 0.000$ | $0.483 \pm 0.000$ | $0.299 \pm 0.000$ | $0.276 \pm 0.000$ |
| Disc Dims | [312, 312] | $0.307 \pm 0.000$ | $0.468 \pm 0.000$ | $0.301 \pm 0.000$ | $0.276 \pm 0.000$ |
| Disc Dims | [128, 128, 128] | $0.300 \pm 0.000$ | $0.471 \pm 0.001$ | $0.295 \pm 0.004$ | $0.253 \pm 0.002$ |
| Disc Dropout | 0.1 | $0.315 \pm 0.000$ | $0.483 \pm 0.000$ | $0.299 \pm 0.000$ | $0.276 \pm 0.000$ |
| Disc Dropout | 0.3 | $0.315 \pm 0.000$ | $0.483 \pm 0.000$ | $0.299 \pm 0.000$ | $0.276 \pm 0.000$ |
| Disc Dropout | 0.2 | $0.312 \pm 0.007$ | $0.480 \pm 0.006$ | $0.296 \pm 0.006$ | $0.273 \pm 0.007$ |
| Disc Dropout | 0.0 | $0.301 \pm 0.002$ | $0.471 \pm 0.001$ | $0.297 \pm 0.002$ | $0.255 \pm 0.007$ |
| GRL $\gamma$ | 5 | $0.310 \pm 0.008$ | $0.479 \pm 0.006$ | $0.296 \pm 0.005$ | $0.270 \pm 0.010$ |
| GRL $\gamma$ | 2 | $0.309 \pm 0.008$ | $0.478 \pm 0.006$ | $0.298 \pm 0.001$ | $0.266 \pm 0.012$ |
| GRL $\gamma$ | 10 | $0.308 \pm 0.007$ | $0.476 \pm 0.006$ | $0.298 \pm 0.002$ | $0.266 \pm 0.012$ |
| Warmup Mult. | 5 | $0.312 \pm 0.007$ | $0.480 \pm 0.005$ | $0.299 \pm 0.001$ | $0.271 \pm 0.010$ |
| Warmup Mult. | 1 | $0.309 \pm 0.008$ | $0.478 \pm 0.006$ | $0.298 \pm 0.001$ | $0.267 \pm 0.012$ |
| Warmup Mult. | 3 | $0.306 \pm 0.007$ | $0.474 \pm 0.006$ | $0.297 \pm 0.005$ | $0.264 \pm 0.011$ |

[*]Disc = Discriminator

### D.3 HYPERPARAMETERS

Table 7 lists the key hyperparameters used for training Geo-DANN DeepScore.

### D.4 CALIBRATION ANALYSIS

We also examine calibration using Brier scores, defined as the mean squared error between predicted conversion probabilities and observed binary outcomes. Our training pipeline logs root-mean-squared Brier (RMSE Brier) per model and region; Table 8 reports the corresponding Brier scores obtained by squaring these values.

Overall calibration is comparable across the three neural models. Multi-Head DeepScore achieves the lowest overall Brier score, while Geo-DANN maintains similar calibration in Americas and East-Asia and slightly improves calibration in Europe relative to Single-Head. Thus, Geo-DANN's gains in geo-fairness (AUPR and Lift@30% gaps) do not come at the cost of degraded probability calibration.

Table 7: Key hyperparameters for DeepScore and Geo-DANN.

| Hyperparameter | Value |
|---|---|
| Transformer layers $L$ | 3 |
| Attention heads $H$ | 8 |
| $d_{\mathrm{model}}$ | 312 |
| Transformer FFN dim | [128, 128] |
| Dropout | 0.1 |
| Weight Decay | 0.0001 |
| Sequence length $T$ | 100 |
| Batch size | 64 |
| Optimizer | AdamW |
| Betas | [0.9, 0.99] |
| Initial learning rate | 0.0001 |
| Scheduler | cosine_with_restarts |
| Scheduler Restarts | 2 |
| Warmup steps | 10 |
| GRL $\gamma$ | 10 |
| Disc dims | [312, 312] |
| Disc dropout | 0.0 |
| Semantic Model | Qwen2-1.5B-instruct |
| Epochs | 50 |
| Max Gradient Norm | 1.0 |
| Feature Embedding Dim | 16 |

Table 8: Brier scores (lower is better) on the held-out test set. Values are obtained by squaring the reported root-mean-squared Brier scores for each model/region.

| Model | Overall | Americas | Europe | East-Asia |
|---|---|---|---|---|
| Single-Head | 0.231 | 0.245 | 0.181 | 0.199 |
| Multi-Head | 0.219 | 0.229 | 0.180 | 0.187 |
| Geo-DANN | 0.223 | 0.233 | 0.178 | 0.187 |

# E  DEEPSCORE ARCHITECTURE DETAILS

## E.1  DEEPSCORE BACKBONE

DeepScore turns the full marketing and sales history of a lead into one long sequence and feeds it to a Transformer encoder. Every interaction (*e.g.* e-mail open, ad click, webinar attendance, phone call) is first mapped to a *learned touch embedding*. This is done through learning feature level embeddings for the various metadata attributes for a touch, and encoding the textual context into a semantic embedding. These feature embeddings, semantic embeddings, and numerical values are concatenated and fed into a linear reduction layer, producing the *learned touch embedding*. Alongside those touch tokens, we concatenate four *discrete time embeddings* (year, month, day-of-month, weekday) so the model can reason about seasonality and working-day effects; borrowing from Lim et al. (2021b). We also adapt the bucketed bias introduced for T5 in to encode relative position because relative distances are more descriptive with very large sequences Shaw et al. (2018). The result is a sequence of lead interactions in the following shape: [ $\underbrace{\text{Touch}}_{\text{embedding}}$ , $\underbrace{\text{Time}}_{\text{embeddings}}$ , $\underbrace{\text{RelPos}}_{\text{bias}}$ ]$_1$ [ Touch, Time, RelPos ]$_2$ . . .

We then learn a representation for static "profile" information about the lead, using 241 categorical and 37 numerical features. This is handled in a similar way to the touch embeddings, where we learn feature-level embeddings for the categorical features, concatenate them with numerical features and reduce them into a profile embedding. We concatenate the lead interaction sequence with the profile information using a separator token and then front pad each sequence to meet a fixed distance $T =$

1536. This information is fed into a transformer encoder with 3 layers and 8 attention heads, following Vaswani et al. (2017). The sequence representation is: $[\;\underbrace{\texttt{pad}}_{\times T-k}\;,\;\underbrace{\texttt{touch/time/pos}}_{\times k}\;,\;\texttt{[SEP]},\;\underbrace{\texttt{profile}}_{1}\;]$

The encoder produces hidden states $\texttt{hidden} \in \mathbb{R}^{T \times 312}$. As in BERT(Devlin et al., 2019), the penultimate position is reserved as a *[CLS]*-style token that summarizes the sequence; the model selects that vector and treats it as the dense *lead representation* $f_\theta(x)$. A single linear layer then converts $f_\theta(x)$ into the logit of conversion, yielding the backbone's binary prediction. Figures 4–8 provide the complete architectural specification with all layer dimensions and connection patterns.

## F  DETAILED ARCHITECTURE

Figure 4 provides the complete architectural blueprint of the Geo-DANN DeepScore model. The architecture consists of: (1) Six parallel embedding modules processing different categorical features (embedding depth 3 with various output dimensions), (2) Three separate embedding branches for touch/time/position information feeding into concatenation layers, (3) Layer normalization and dropout regularization at multiple stages, (4) The core transformer encoder with positional encoding, (5) Parallel paths for the gradient-reversal geography discriminator and geo-specific task heads, and (6) Final classification layers with domain discriminator and cross-entropy loss computation. All tensor shapes are preserved throughout the forward pass, enabling end-to-end gradient flow for both the primary task and adversarial objectives.

### F.1  COMPONENT-LEVEL ARCHITECTURAL DETAILS

To facilitate precise implementation and ablation studies, we provide detailed diagrams of the four key architectural components. These diagrams show exact tensor dimensions at each layer, enabling reproduction of our results.

**Embedding Processing (Figure 5):** The touch embedding pipeline processes heterogeneous input features by first encoding textual context using the Qwen2-1.5B-instruct model, producing 1536-dimensional embeddings. Three categorical features are embedded separately into 16-dimensional vectors each. These components are concatenated (1536 + 3X16 = 1584 dimensions) and then projected to 312 dimensions through a learned linear transformation. A parallel ProfileEmbeddings module (shown in Figure 4) handles static lead features using a similar concatenation and projection architecture.

**Transformer Encoder (Figure 6):** Our transformer backbone implements a modified architecture optimized for sequential lead scoring. Each encoder layer contains: (i) Multi-head attention with 8 heads operating on 312-dimensional representations ($d_{model} = 312$), producing context-aware embeddings through scaled dot-product attention; (ii) A position-wise feedforward network with intermediate dimension 128 and dropout rate 0.3 for regularization; (iii) Layer normalization (eps=$10^{-3}$) and residual connections following the standard transformer design (Vaswani et al., 2017). The encoder stack consists of 3 such layers, chosen to balance model capacity with training efficiency on our 1.4M-lead dataset.

**Domain-Adversarial Module (Figure 7):** The DANN component implements adversarial training through a gradient reversal layer (GRL) connected to a domain discriminator. The discriminator is a 3-layer network with two 312X312 hidden layers and a final 312X16 output layer, where the 16-dimensional output matches the feature embedding dimension for binary cross-entropy (BCE) loss computation across geography classes. The network uses GELU activations and no dropout (p=0.0) to ensure stable adversarial training. During forward propagation, the GRL acts as identity; during backpropagation, it multiplies gradients by $-\lambda_{\text{GRL}}$ following the schedule in Equation 3. The BCE loss from geography classification is computed and backpropagated with reversed gradients, forcing the encoder to learn geography-invariant representations.

**Prediction Heads (Figure 8):** The geo-specific prediction module contains 10 lightweight heads (one per geography), each implemented as a single linear layer. Each head processes the 312-dimensional shared representation through dropout (p=0.3) for regularization, then projects directly to scalar logits. This simple design allows each geography to learn modest regional adjustments while relying primarily on the shared geo-invariant representation from the encoder.

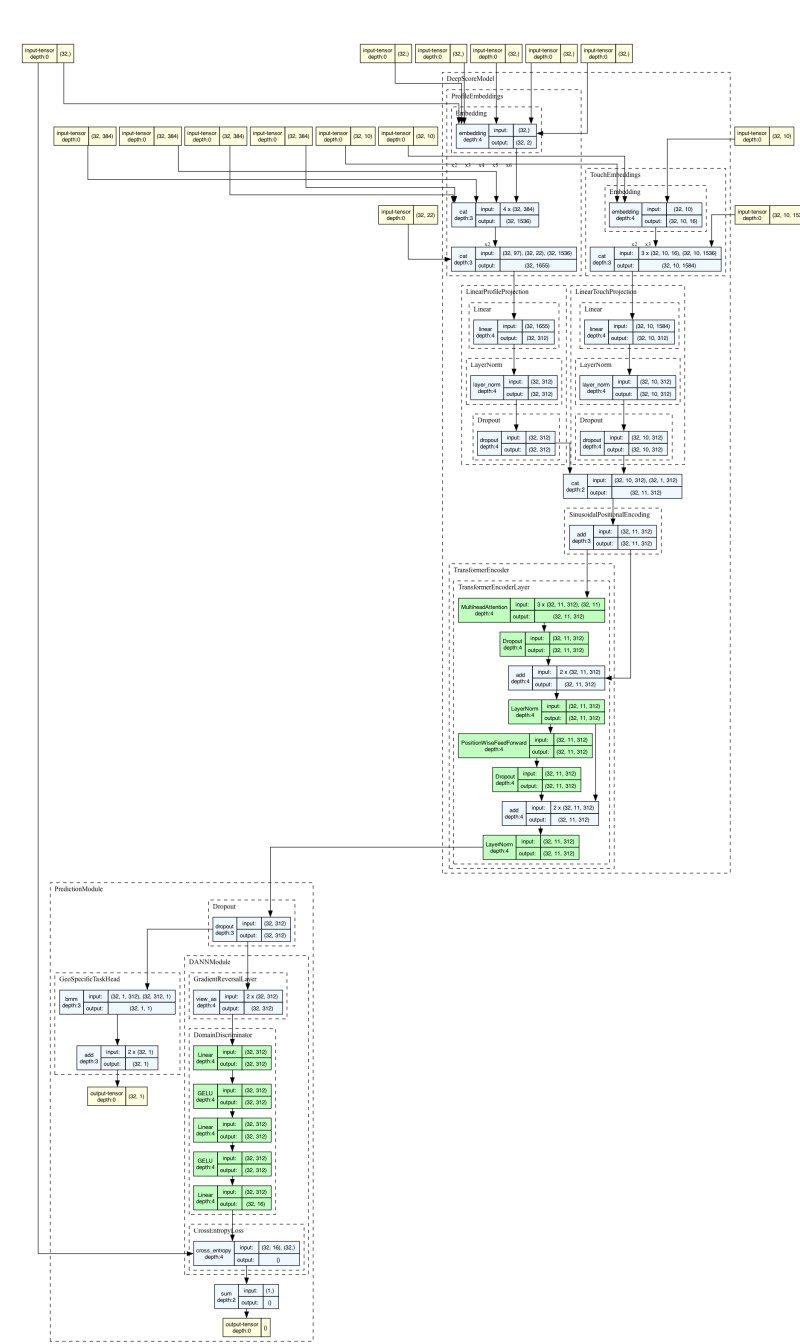

Figure 4: Complete DeepScore architecture with detailed layer specifications. The diagram shows the full information flow from input embeddings through the transformer encoder to the final predictions, with all components annotated with tensor dimensions and layer specifications.

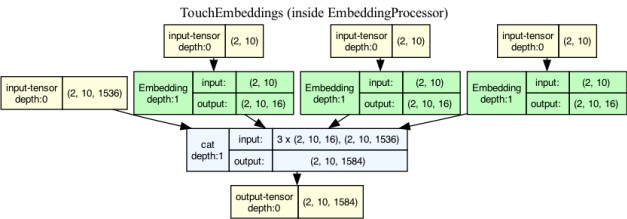

Figure 5: Touch embeddings processing pipeline. Textual context is encoded using Qwen2-1.5B-instruct producing 1536-dimensional embeddings, which are concatenated with three 16-dimensional categorical feature embeddings (total: 1584 dimensions), then projected to 312 dimensions via a linear layer. A similar process handles profile embeddings (see Figure 4).

## F.2 IMPLEMENTATION DETAILS

The complete architecture processes sequences of up to 100 timesteps, with each input token combining learned embeddings for touch interactions, temporal features (4 discrete time embeddings), and relative position encodings. As shown in the component diagrams (Figures 5–8), the information flow proceeds through four distinct stages:

**Stage 1 - Input Processing:** The embedding pipeline (Figure 5) encodes textual context using the Qwen2-1.5B-instruct model to produce 1536-dimensional embeddings. Three categorical features are embedded into 16-dimensional vectors each and concatenated with the text embeddings (1536 + 3X16 = 1584 dimensions total). This concatenated representation is then projected to 312 dimensions through a learned linear transformation.

**Stage 2 - Sequence Modeling:** The transformer encoder (Figure 6) applies self-attention across the full sequence using 8 attention heads operating on 312-dimensional representations ($d_{model} = 312$). Each of the 3 encoder layers includes multi-head attention, position-wise feedforward networks (with intermediate dimension 128), layer normalization (eps=$10^{-3}$), and dropout regularization (p=0.3). This configuration was optimized through extensive hyperparameter search to balance model capacity with training stability on our 1.4M-lead dataset.

**Stage 3 - Adversarial Alignment:** The DANN module (Figure 7) implements geography-invariant learning through a gradient reversal layer feeding a 3-layer discriminator network with architecture [312, 312, 16]. The discriminator uses GELU activations and no dropout (p=0.0) to ensure stable adversarial training. During backpropagation, gradients are multiplied by $-\lambda_{\text{GRL}}$ according to the schedule in Section 3.3, creating an adversarial signal that removes geographic artifacts from the shared representation.

**Stage 4 - Geo-Specific Prediction:** The prediction module (Figure 8) contains 10 lightweight heads, one per geography, each implemented as a single linear layer with dropout (p=0.3) for regularization. These heads learn modest regional adjustments while relying primarily on the shared geo-invariant representation from the encoder.

This modular design enables targeted ablations of individual components while maintaining end-to-end differentiability. The detailed tensor dimensions shown in each component diagram (Figures 5–8) facilitate exact reproduction and guide implementation decisions for practitioners adapting this architecture to related problems.

## G REPRODUCIBILITY STATEMENT

To ensure reproducibility of our results, we provide comprehensive experimental details throughout the paper and appendices. Our dataset covers a two-year window of marketing-qualified leads with a 2.5-month hold-out period, as detailed in Section 5.2. The complete model architecture, including all layer specifications, embedding dimensions, and connection patterns, is fully documented in Appendix F (Figures 4–8). The transformer backbone configuration (3 layers, 312 hidden dimensions, 8 attention heads), attention mechanisms, and DANN components are specified in Section 3 with detailed component-level diagrams in Appendix F. Key hyperparameters for the DANN module

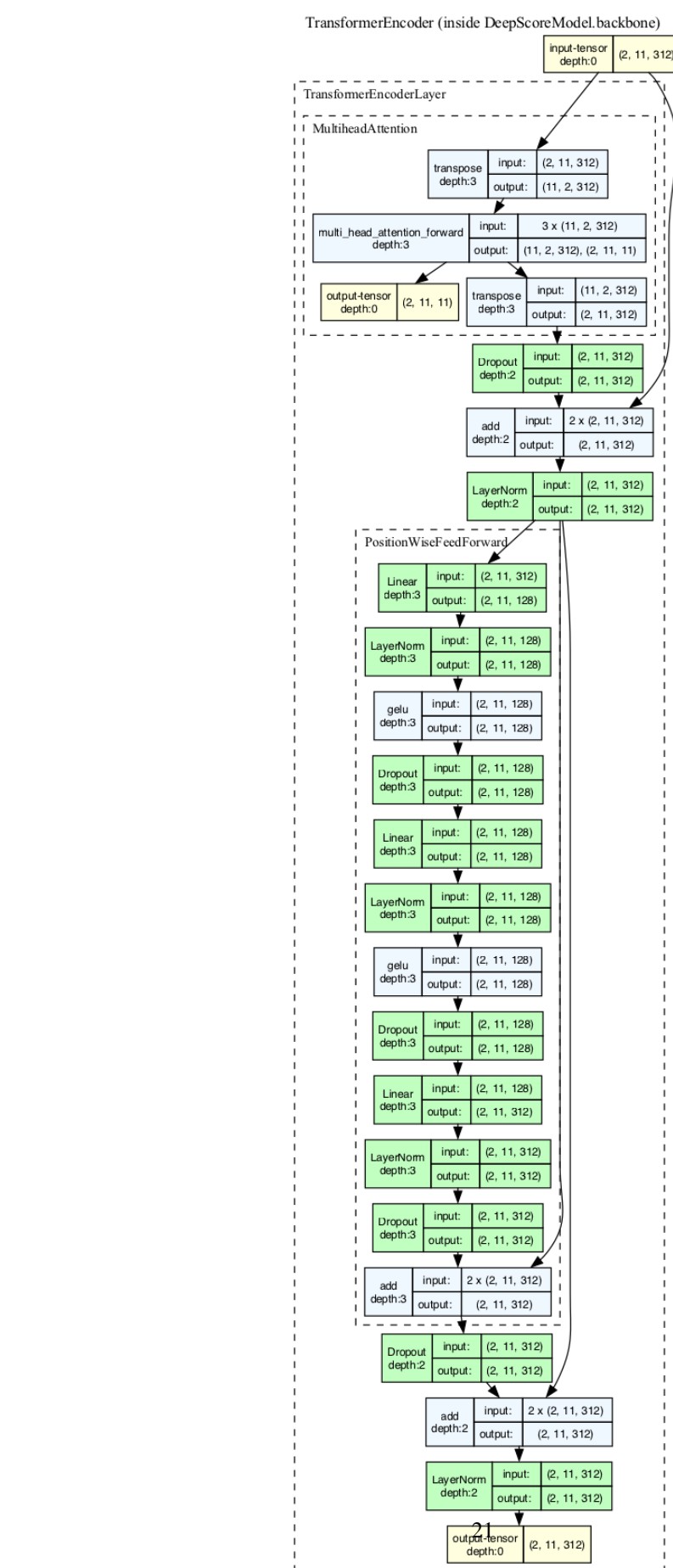

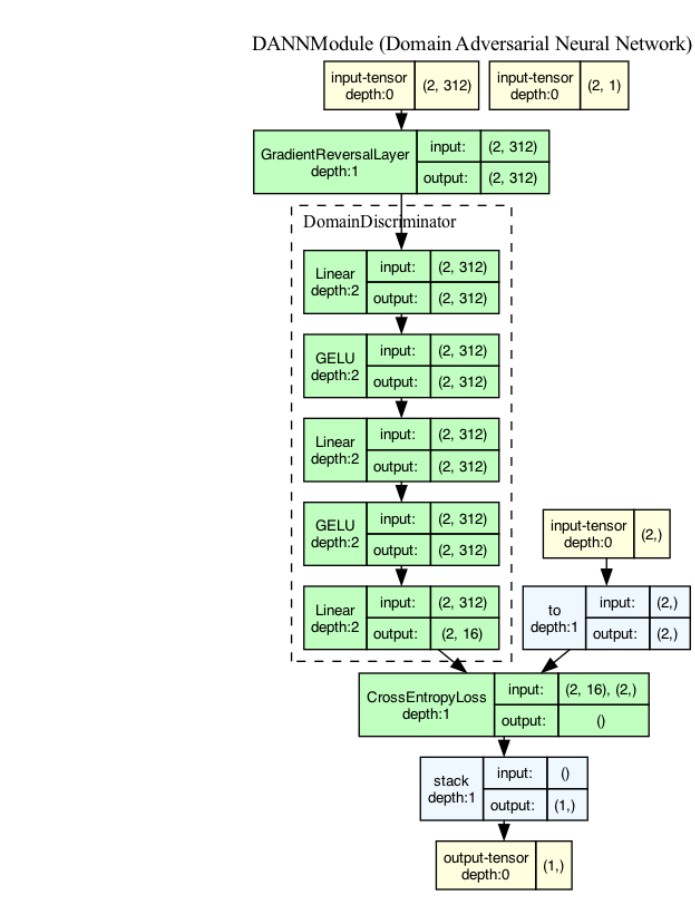

Figure 7: DANN module with gradient reversal. Gradient reversal layer feeding a 3-layer domain discriminator with two 312X312 hidden layers and a final 312X16 output layer (16 matches feature embedding dimension), using GELU activations and no dropout (p=0.0) for stable adversarial training. BCE loss is computed on the 16-dimensional output.

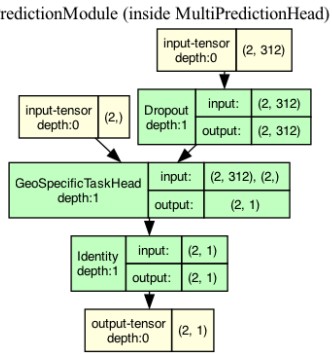

Figure 8: Geo-specific prediction heads. Each of the 10 geographic regions has a lightweight single-layer linear classifier with dropout (p=0.3) that projects the 312-dimensional shared representation to scalar logits.

include the gradient reversal scheduling ($\lambda_{\mathrm{GRL}}$ and $\gamma$) described in Section 3.3 and tabulated in Table 7. All experiments were conducted with AdamW optimizer. For model training, we used a fixed sequence length $T = 1536$, Qwen2-1.5B-instruct for text embeddings, and the architectural specifications shown in Figures 5–8. Performance metrics (AUPR and Conversion-Rate Lift@30%) are clearly defined in Section 5.2. All reported results are averaged across multiple hyperparameter optimization runs to account for stochastic variations. While we cannot share proprietary business data, our methodology section and architectural diagrams provide sufficient detail for implementation on similar B2B lead scoring datasets.

## H  DATA AND TEMPORAL SPLIT DETAILS

To enable a like–for–like comparison with the production LIGHTGBM baseline, every DeepScore variant is trained on the *same* two-year window of marketing-qualified leads (May 2022–May 2024). The corpus comprises ten regional business units and 1.4M labeled examples. Generalization is assessed on the subsequent 2.5-month hold-out period (Jul 2024–Sep 2024), a horizon long enough to capture genuine market drift yet short enough to respect the median 60-day qualification-to-opportunity lag.

We evaluate all models with two complementary metrics: average precision–recall (AUPR), which summarizes ranking quality under extreme class imbalance, and Conversion-Rate Lift@30%, which measures relative conversion rate within the top-ranked 30% of leads and reflects how quota-constrained sales teams triage prospects in practice. We report macro-AUPR and per-geo AUPR and Lift@30% to quantify both overall performance and geographic disparities. Our temporal split uses May 2022-May 2024 for training and July-September 2024 for testing, with a one-month gap to respect the median 60-day qualification-to- opportunity lag. This design captures realistic business drift while avoiding label leakage; due to internal constraints we do not evaluate on multiple future time windows and leave a broader temporal robustness analysis to future work.

For each lead we truncate the interaction sequence at the qualification time and compute features only from events observed up to that time; any post-decision events are excluded from the input. Each lead appears in exactly one of {train, validation, test}, ensuring that information does not leak across splits. The positive class (conversion) is rare, with an overall prevalence in the low single digits. Regional prevalences are similar, with the highest-prevalence geography having less than $1.2\times$ the prevalence of the lowest-prevalence geography. Due to business confidentiality, we cannot report exact per-region base rates or normalized AUPR values, but our primary metrics—macro-AUPR and per-geo AUPR and Lift@30%—are unaffected by this and allow us to quantify both overall performance and cross-geo disparities.

## I  MULTI-HEAD GEO-SPECIFIC RESIDUALS

Complete invariance is not desirable when conditional distributions $P(Y \mid X, G)$ differ across geographies. Following Zhao et al. (2019), we address this by attaching a small classifier $C_{\phi_g}$ for each geography $g$ on top of the shared encoder. After the encoder has reduced cross-geo divergence, these heads model residual differences without re-introducing large divergence. This provides a compact interpretation of our empirical findings: Geo-DANN reduces the divergence term while geo-specific heads keep the joint-error term $\lambda^*$ small, improving under-represented regions without hurting the majority domain.

## J  THE USE OF LARGE LANGUAGE MODELS (LLMS)

Large Language Models (LLMs) were utilized in multiple capacities during the preparation of this research. Specifically, we employed LLMs as research assistants to aid in literature discovery and background research. This involved using LLM-powered tools to search for relevant prior work, generate keyword lists for comprehensive literature searches, and create comparison tables of related studies. These tools helped us efficiently navigate the vast landscape of scientific literature, ensuring a thorough and up-to-date background section. Additionally, LLMs were used as writing assistance tools to improve clarity, suggest alternative phrasings, and refine grammar. However, all scientific contributions, including the core idea of applying DANN to B2B lead scoring, theoretical analyses,

architecture design, experimental methodology, and result interpretations, are original work conceived and developed by the authors. No LLMs were used for research ideation, experimental design, or data analysis. All technical claims and empirical results were independently verified through rigorous experimentation. We take full responsibility for the paper's contents, having thoroughly fact-checked all statements, including those refined with LLM assistance. The scientific novelty and intellectual contributions of this work are entirely attributable to the human authors listed.

