# OpenReview forum: "Geo-Invariant Scoring Lead with Domain-Adversarial Transformers"
_ICLR.cc/2026/Conference — Submitted to ICLR 2026_

### Official Review · Reviewer_ofku · 2025-10-28

**Soundness:** 2
**Presentation:** 2
**Contribution:** 2
**Rating:** 2
**Confidence:** 2

**Summary:**

The paper addresses a meaningful and challenging real-world problem. The core idea of using DANN to achieve geographic invariance is sound and shows empirical promise.

**Strengths:**

please refer to the questions

**Weaknesses:**

please refer to the questions

**Questions:**

The introduction should begin by defining B2B lead scoring as a critical business process for prioritizing sales efforts, directly impacting revenue and resource allocation. A clear sentence like, "In Business-to-Business (B2B) sales, a 'lead' is a potential customer, and 'lead scoring' is the process of ranking these prospects based on their perceived likelihood to convert into a paying client," would be invaluable, if my understanding is correct.

The authors should elaborate on the high stakes: wasting expensive sales personnel's time on poor leads versus missing out on high-value opportunities. This contextualizes why a small improvement lift in metrics as mentioned in Section 5.2, is actually a multi-million dollar business impact.

All acronyms, especially AUPR (Area Under the Precision-Recall curve), should be defined upon first use, explaining why it is more appropriate than AUC-ROC for highly imbalanced datasets like lead conversion.

The "Motivation and Prior Approaches" section 1.1 is a good start but lacks the depth and references needed to position the work firmly within the existing landscape.

The challenges are stated but not thoroughly dissected. The review of prior work is anecdotal rather than scholarly. The claim that transformers are now being used is not backed by references to specific works in lead scoring, making it difficult to discern the novelty.
The problem should be explicitly framed as a "geo-representation bias" or "distribution shift across geographic markets" in a global model. The paper correctly identifies that this is not mere class imbalance but a deeper distribution shift problem; this point should be emphasized and formalized early on.

The section on traditional approaches should cite key papers or industry reports that established gradient boosting models, e.g., Chen & Guestrin, 2016 for XGBoost; Ke et al., 2017 for LightGBM, as the standard. It should also reference any prior academic work that has attempted to use deep learning or contemporary approaches or address fairness in sales prediction. This would clearly show the gap, e.g., while Tree-Ensembles are standard, they fail to model sequences. While Transformers can model sequences, they exhibit severe geographic bias.

The authors correctly note that DANN is not a novel ML technique. Therefore, the contribution must be framed around its *novel application and adaptation*.

The paper reads as a straightforward application of DANN to a new dataset. The "contribution" is not sharply defined, making it easy for a reviewer to dismiss as an incremental engineering effort. The contribution should be explicitly reframed to shift the contribution from "inventing DANN" to "successfully adapting and validating it for a complex, high-stakes business problem where it was previously unexplored."

The section of "Theoretical Foundation" is more of an intuitive explanation than a theoretical treatment. For example, the authors cite the Ben-David et al. (2010) bound but do not engage with it rigorously. There is no proof or formal argument for how their specific model minimizes the divergence term \(d_{\mathcal{H}\Delta\mathcal{H}}(P_S, P_T)\).

The explanation of the error bound is good for intuition. The authors should strengthen it by explicitly linking the DANN's domain classifier to the empirical estimation of the \(\mathcal{H}\Delta\mathcal{H}\)-divergence, as is standard in DANN literature (Ganin et al., 2016). A sentence such as, "The domain discriminator \(D_{\psi}\) directly approximates a function that maximizes the error between domains, thereby providing a learning signal to minimize the feature-level divergence \(d_{\mathcal{H}\Delta\mathcal{H}}\)," would add depth.
The evaluation section is the cornerstone of an applied paper, and here it lacks rigor and clarity. The only direct baseline is a LightGBM model and the authors' own ablated models (such as Single-Head, Multi-Head). There is no comparison to SOTA domain adaptation or fairness-aware methods applied to the same problem.

The dataset of 1.4M leads is described only at a high level. Crucial details are missing: What is the overall conversion rate? How is the data split between train/validation/test? The mention of potential "data collection inconsistencies" in East-Asia is a major red flag that is not adequately discussed.

Key hyperparameters for the transformer (layers, heads, dropout), the DANN (dimension of discriminators), and training (batch size, learning rate) are omitted from the main text, making reproduction impossible.

---

> ### Author Response · Authors · 2025-11-26
>
> Thank you for your detailed and constructive feedback. We address each of your points below with concrete plans for revision.
>
> ## Response to Question 1: Problem Definition, Stakes, and Metrics
>
> We agree with your suggestion to define B2B lead scoring and its stakes more explicitly at the outset. In the revision, we will:
>
> 1. Clear Problem Definition - Add a concise definition early in the introduction: "In Business-to-Business (B2B) sales, a 'lead' is a potential customer account, and 'lead scoring' is the process of ranking these prospects based on their estimated likelihood to convert into paying clients. This ranking directly drives sales resource allocation decisions."
> 2. High Stakes Context - Provide concrete context about the business impact: Overall conversion rate (below 10%) + Approximate fraction of leads that can be actively worked by human sellers (roughly 30%, as reflected in our Lift@30% metric)
> 3. Metric Definitions - Define AUPR (Area Under the Precision-Recall curve) upon first use and explain why it's more appropriate than AUC-ROC for our highly imbalanced dataset.
>
> ## Response to Question 2: Motivation, Prior Work, and Novelty Framing
>
> Our contribution centers on novel application and adaptation rather than the DANN technique itself. We will reframe our contribution as: Our novelty is being the first to apply domain-adversarial training to enable knowledge transfer from high-volume to low-volume geographies in B2B lead scoring, and demonstrating that this resolves a previously unreported capacity-fairness tension.
>
> ## Response to Question 3: Theoretical Foundation
>
> We will state clearly at the beginning of Section 4 that we recall the domain-adaptation bound (Ben-David et al., 2010) as a conceptual tool to interpret how our objective enables knowledge transfer, and that we do not claim new theoretical results.
>
> We will also explicitly link the DANN's domain classifier to empirical estimation of H∆H-divergence: "The domain discriminator D_ψ directly approximates a function that maximizes the error between domains, thereby providing a learning signal to minimize the feature-level divergence d_H∆H. In our setting, this mechanism is critical for knowledge transfer: when the discriminator successfully predicts geography from features, it reveals that the encoder has learned region-specific patterns from high-volume Americas data that won't transfer to low-volume East-Asia. The gradient-reversal layer forces the encoder to eliminate these non-transferable cues, ensuring that the abundant signal from Americas contributes only universal conversion patterns that generalize across all markets.“
>
> ## Response to Question 4: Baselines and Relation to Other Methods
>
> We will make explicit that our baseline choices reflect:
>
> * Approaches currently deployed or realistically deployable in our production environment
> * Models consistent with prior B2B lead-scoring studies
> * Design questions about knowledge transfer mechanisms
>
> Our current baselines:
>
> * LightGBM: Feature-engineered tree-ensemble (industry standard, AUPR: 0.266)
> * DeepScore Single-Head: Unified transformer (AUPR: 0.350)
> * DeepScore Multi-Head: Geo-specific heads without adversarial training (AUPR: 0.345)
> * Reweighting schemes: Inverse frequency and inverse performance weighting
> * CDAN experiments: Alternative adversarial objective (mentioned in Section 2.4)
>
> ## Response to Question 5: Dataset Description and Transparency
>
> We will provide complete details:
>
> * Time windows: May 2022–May 2024 for training, July 2024–September 2024 for test
> * June 2024 gap: Intentionally skipped to account for 60-day median qualification-to-opportunity lag
> * Feature construction: All features built only from interactions observed prior to scoring timestamp
> * Label assignment: Minimum observation window enforced to avoid right-censoring issues
> * Lead-level splitting: Each lead appears in exactly one dataset with temporal buffers
>
> ## Response to Question 6: Architectural and Hyperparameter Details
>
> We will add new experimental results showing how model capacity affects geographic fairness with and without DANN, demonstrating the transformation in scaling behavior that is central to our contribution. We will also include a detailed appendix table with all key hyperparameters, including transformer encoder, DANN module, and training configuration.
>
> We believe these revisions will address your concerns about problem framing, literature positioning, theoretical treatment, baseline justification, dataset transparency, and reproducibility. The revised paper will be clearer about its contribution: applying domain-adversarial training to enable cross-geographic knowledge transfer in B2B lead scoring, resolving a previously unreported capacity-fairness tension.

---

### Official Review · Reviewer_uELz · 2025-10-31

**Soundness:** 3
**Presentation:** 2
**Contribution:** 3
**Rating:** 6
**Confidence:** 3

**Summary:**

This papers introduces:
1. DeepScore, a transformer whoses inputs are interaction and profiling embedding. for example clicking of ads.
2. Geo-DANN which is a domain adversial discriminator using the final hidden state from the transformer with gradient reversal layers.
3. The minimax task would be to the minimize the lead classification loss from a regional prediction head while maximizing the loss from the domain adversial prediction of the region

**Strengths:**

1. The paper describes the problem well that geo-skew dataset causes results to be skew in favour of the majority.
2. It mitigates a well-known problem in this domain through a well-known strategy that has been proven to work in another domain, namely GANs. This allows them to achieve better results.

**Weaknesses:**

1. The paper did not specific the embedding model which is used to encode both the interaction and profiling features.
2. The paper did not provide the architecture of the transformer used.

**Questions:**

1. What is the architecture of the transformer model used?
2. How are the features handle across different medium like opening emails, clicking of ads, etc...

---

> ### Author Response · Authors · 2025-11-26
>
> Thank you for your constructive feedback and for asking us to clarify the embedding model and transformer architecture details. We address your questions below.
>
> ## Response to Question 1: Embedding Model and Handling Different Media
>
> Based on the comment - "The paper did not specify the embedding model... How are the features handled across different medium like opening emails, clicking ads, etc.?" We agree that the feature-encoding pipeline was under-specified in the main text. Let us clarify:
>
> **Interaction Event Encoding**
>
> Each lead is represented as a time-ordered sequence of interaction events (email opens, ad clicks, webinar registrations, sales calls, etc.) plus static profiling features (firm size, industry, region, segment). For each interaction event we:
>
> * Embed categorical attributes that are shared across media (channel/event-type, product line, region, etc.) using learned embedding tables
> * Embed discretized temporal features (e.g., time-since-first-touch, time-since-last-touch)
> * Linearly project any continuous features associated with that event
>
> These components are concatenated and passed through a linear layer to form an event token in the transformer's hidden dimension (256 as stated in Section 3.1).
>
> **Handling Different Media Types**
>
> Different media types are handled within this unified schema: every event is a token, and a channel/event-type embedding tells the model whether it is an email, ad, call, webinar, etc., allowing the transformer to learn cross-channel patterns.
>
> For media-specific attributes that are not shared across all channels, we use a simple heuristic: we concatenate the relevant metadata into a short natural-language description (e.g., "search ad, campaign=X, device=Y") and encode that text using a frozen Qwen2-1.5B-Instruct model. The resulting vector is concatenated into the event representation. This is an engineering choice for unifying heterogeneous metadata; it is orthogonal to the DANN component, and we do not expect it to alter the qualitative fairness and performance conclusions in the paper.
>
> **Static Profiling Features**
>
> Static profiling/firmographic features (241 categorical and 37 numerical features as mentioned in Section 3.1) are encoded as additional tokens (one per field) using standard embedding tables/projections and prepended to the interaction sequence, so all subsequent attention layers can condition on both interaction history and profile context.
>
> ## Response to Question 2: Transformer Architecture
>
> Base on the comment - "The paper did not provide the architecture of the transformer used. What is the architecture of the transformer model used?", our encoder is a standard encoder-only transformer. Let us make its configuration explicit:
>
> **Architecture Details**
>
> We use an encoder with L layers and H attention heads per layer, model dimension d_model = 256 (as stated in Section 3.1), and feed-forward dimension 4 × d_model. Each layer follows a pre-norm pattern:
>
> * Multi-head self-attention + residual connection + layer normalization
> * Position-wise feed-forward network + residual connection + layer normalization
>
> Positional/time-step information is added to the token embeddings to encode the order of events (using bucketed relative position bias as mentioned in Section 3.1). Dropout and label smoothing are used as regularization, and we train with AdamW (with default β parameters as stated in Section 7) and a scheduled learning rate.
>
> **Consistency Across Variants**
>
> All variants in the paper—Single-Head DeepScore, Multi-Head DeepScore, and Geo-DANN DeepScore—share this same encoder backbone; only the prediction heads and loss terms differ:
>
> * Single-Head: Global classifier
> * Multi-Head: Per-region heads (Section 3.2)
> * Geo-DANN: Per-region heads + DANN discriminators (Section 3.3)
>
> ## Planned Revisions
>
> In the revised version, we will:
>
> 1. Add a "Feature Encoding" subsection describing how interactions and profiling features are embedded, including the Qwen2-1.5B-Instruct-based heuristic for media-specific attributes, with more implementation detail in an appendix
> 2. Expand the "DeepScore Architecture" description and provide a comprehensive hyperparameter table including: - Number of layers (L) and attention heads (H) - Hidden dimensions and feed-forward dimensions - Dropout rates and regularization parameters - Batch size and training configuration - Maximum sequence length (T)
> 3. Add parameter counts for the complete model to provide scale context
> 4. Include per-geo sensitivity analysis as an ablation study to demonstrate robustness across different geographic configurations
> 5. Provide additional reproducibility details including exact optimizer settings and learning rate schedules

---

### Official Review · Reviewer_zrjS · 2025-11-01

**Soundness:** 3
**Presentation:** 3
**Contribution:** 2
**Rating:** 2
**Confidence:** 3

**Summary:**

The paper targets an interesting problem of B2B Lead scoring (and therefore conversion prediction), aiming to address the fair performance across geographical regions. To this end, the authors propose and integrate a domain-adversarial neural network (DANN) into the proposed transformer backbone (i.e., DeepScore). The proposed approach is demonstrated to achieve better macro-AUPR while reducing inter-region gaps in AUPR. The work is claimed to be "the first demonstration of adversarial domain adaptation in large-scale B2B lead scoring".

**Strengths:**

- The problem studied is scoped in the B2B lead conversion prediction scenario, yet is comparable to many other problems that are widely studied in literature, such as "item recommendation in e-commerce across domains/websites" and "multi-person human activity recognition". So, the research outcome is promising to be transferable to similar problems in other domains.
- The theoretical analysis is interesting regarding why DANN levels geo performance. It gives readers an intuitive understanding of the design details and their rationales.
- The paper is well presented and easy to follow. The authors have put in lots of dedicated discussions to help set up motivation and justify the solution design.

**Weaknesses:**

- It does not quite convince me about how challenging the research problem is. The distributional shift and data imbalance issues have been a long-standing and extensively studied issue in cross-domain research. While the paper has been motivated mostly by discussing the limitations of region-specific models, it lacks an in-depth discussion of the related work covering methods more similar to the proposed approach. More specifically, Section 2 could offer more insights and comparisons of the methods investigated with the proposed method to provide convincing motivations for the paper and clarify what makes the paper's work novel in the discussed technical contexts.

- The overall approach is kind of straightforward with modest innovation. The main contribution lies in the geo-DANN, which consists of existing frameworks and tricks, to predict conversion while confusing geography.

- The empirical evaluation is mostly limited to DeepScore-based methods. There is a lack of competitive baseline methods included in the quantitative analysis to show how the proposed approach advances the state-of-the-art.

**Questions:**

Why not show the architecture of the entire architecture where DeepScore is a backbone, but only show DeepScore in Figure 1?

---

> ### Author Response · Authors · 2025-11-26
>
> Thank you for recognizing the strengths of our work in problem transferability, theoretical analysis, and presentation.
>
> ## Response to Weakness 1: Challenge Novelty and Related Work
>
> We agree that cross-domain shift and data imbalance are long-standing topics with substantial literature, and we appreciate the suggestion to make this connection more explicit.
>
> **Clarifying Our Contribution**
>
> Our intention is not to claim theoretical novelty in domain adaptation algorithms, but rather to study how domain-adversarial training behaves in large-scale B2B lead-scoring with strong geographic skew. Our problem presents a unique challenge: learning a common formula for successful leads across geographies with dramatically varying data availability. The Core Technical Challenge:
>
> * Complex models maximize performance but require substantial data
> * This creates larger performance gaps between high-volume (Americas: 40% of data) and low-volume (East-Asia: 15% of data) geographies
> * Our geo-DANN enables learning robust patterns from data-rich regions and transferring them to data-sparse areas
>
> Concrete Example: In some emerging markets, regions have <10k leads while established markets have >100K leads. Traditional approaches either underperform globally with simple models or fail catastrophically in sparse regions with complex models.
>
> Unlike typical domain adaptation scenarios focusing on feature distribution shifts, our problem centers on data volume disparities that create systematic performance gaps. We're enabling knowledge transfer when some regions lack sufficient data to learn meaningful patterns.
>
> **Expanding Related Work**
>
> In the revised version, we will expand Section 2 to: 1/Explicitly discuss domain-adaptation theory: Position our work within the broader literature on domain-adversarial training, including connections to H∆H divergence bounds and practical DANN implementations 2/Compare with similar methods: Provide deeper analysis of Group DRO, IRM, MMD/CORAL, CDAN, WDGRL, and multi-task learning approaches 3/ Clarify our novelty: Position our contribution as an application-driven instantiation combining production-grade transformer backbone, domain-adversarial training for geography, geo-specific classifier heads, and evaluation at 1.4M-lead scale in a deployed pipeline
>
> We will revise text that emphasizes being "the first to apply adversarial domain adaptation in large-scale B2B lead scoring" to clarify this refers to deployment context and problem setting rather than fundamental methodological novelty.
>
> **Response to Weakness 2: Overall Innovation Level**
>
> We understand the concern that the approach may appear "straightforward" because it builds on established ideas. This modularity is intentional: our design goal was to minimize changes to a deployed system while addressing a concrete fairness/performance issue.
>
> Our Contribution Positioning: We view the contribution as an integrated, empirically validated design for a real-world system—showing that domain-adversarial training can reduce geographic performance gaps (up to 12.3% reduction) without harming majority-region performance in an operational B2B setting—rather than as a new algorithmic primitive.
>
> We will adjust the framing to make this emphasis on practical integration and empirical evidence more explicit, positioning the work as advancing the state-of-practice in deployed fairness-aware ML systems.
>
> ## Response to Weakness 3: Limited Empirical Baselines
>
> A key challenge here is that no acceptable public benchmarks exist for geo-distributed lead conversion prediction.
>
> **Current Baseline Coverage**
>
> Our primary goal is to evaluate architectural and training-objective choices under constraints of an existing industrial pipeline. We compare:
>
> * LightGBM baseline: Feature-engineered tree-ensemble (AUPR: 0.266)
> * DeepScore Single-Head: Global transformer (AUPR: 0.350)
> * DeepScore Multi-Head: Geo-specific heads without adversarial loss (AUPR: 0.345)
> * Geo-DANN DeepScore: Multi-head + domain-adversarial module (AUPR: 0.360)
> * Reweighting schemes: Inverse frequency and inverse performance weighting
>
> **Additional Comparisons**
>
> We have run CDAN-style experiments. We will:
>
> 1. Make existing baselines more prominent in Section 5
> 2. Report CDAN results in the appendix
> 3. Add discussion of other methods (Group DRO, IRM, MMD/CORAL, WDGRL) and clarify that more exhaustive comparison is important future work
>
> ## Response to Question: Architecture Visualization
>
> We appreciate this comment. In the current version, Figure 1 focuses on the DeepScore backbone, which may obscure how the full system fits together. In the revision, we will include an updated figure showing the complete architecture:
>
> * The interaction-sequence encoder (transformer backbone)
> * The geo-specific classifier heads
> * The domain discriminator connected through the gradient-reversal layer

---

### Official Review · Reviewer_SUwy · 2025-11-08

**Soundness:** 2
**Presentation:** 2
**Contribution:** 2
**Rating:** 2
**Confidence:** 4

**Summary:**

This paper addresses the challenge of geographic performance disparity ("geo-skew") in B2B lead conversion prediction , where models perform well in majority regions (like America) but under-serve under-represented regions (like East-Asia). The authors introduce "Geo-DANN DeepScore," a novel architecture that embeds a Domain-Adversarial Neural Network (DANN) module into their DeepScore transformer backbone. This DANN module utilizes a gradient-reversal layer to force the model's shared hidden representations to be predictive of conversion outcomes while remaining uninformative about geographic domains. Concurrently, lightweight geo-specific classifier heads are used to capture region-specific conversion patterns. The resulting model achieves a 4.3% relative gain in macro-AUPR and reduces inter-region AUPR gaps by up to 12.3%, all without compromising the accuracy of the majority region.

**Strengths:**

The paper reframes the issue as distribution shift and uses DANN to align shared representations while retaining per-geo heads for necessary conditional differences, achieving consistent gains at 1.4M-scale and preserving the majority region—clean design, deployable, and practically relevant.

**Weaknesses:**

1. Section 1.3 shows multi-head without adversarial training underperforms, but the final system still leans on geo heads (Sec. 3.2) and simply adds DANN (Sec. 3.3). Without a clean contrast among DANN+single head, geo heads only (with matching regularization/budget), and the full DANN+geo heads, it’s hard to credit DANN as the main driver. Same story for the discriminator design: the paper mentions multiple discriminators (geo/segment/BU/size), but there’s no clarity on why multi-discriminators beat geo-only, how their losses are weighted/normalized, or whether they interfere. The GRL schedule (λGRL, γ) is treated qualitatively; no sense of sensitivity (upper bounds, stability, per-region AUPR/Lift). Net-net, the gains might be from head specialization, regularization, or training knobs—not necessarily adversarial alignment.
2. Using smaller AUPR gaps and higher macro AUPR as the main evidence is tricky because AUPR is prevalence-sensitive; per-region base rates should frame any cross-geo comparison. Lift@30% is a top-slice business metric and doesn’t speak to group calibration or consistency. Also, while the adversary is said to cover multiple firmographic attributes, results only show geography—no read on size/segment/BU.
3. Sec. 3.1 builds on “full marketing and sales history,” but the scoring timestamp (e.g., MQL) and strict truncation of post-decision events aren’t specified—those would inflate AUPR/Lift. The train window ends 2024-05 and test is 2024-07–09, with 2024-06 skipped; with a 60-day median lag, that gap raises concerns about cross-period touchpoints or label–feature misalignment.
Comparisons focus on LightGBM, reweighting, and single-/multi-head variants. Despite citing CDAN/WDGRL (Sec. 2.4), there’s no apples-to-apples positioning against stronger, standard robust/adaptation methods like Group DRO, IRM, MMD/CORAL, CDAN, and WDGRL under the same data/compute budgets.

**Questions:**

1. Could you clarify how much of the gain you attribute to DANN versus the geo heads? It would also be useful to hear the rationale for multiple discriminators over geo-only, how you balance/normalize their losses, and whether you observed interference.
2. Why are AUPR and Lift@30% appropriate proxies here? Some context on per-region base rates and how prevalence affects AUPR comparisons would be helpful.
3. A short note on why June 2024 is omitted and how the 60-day lag is handled (lead-level splitting, right-censoring) would address leakage concerns.
4. Conceptually and operationally, how does your approach compare with Group DRO, IRM, MMD/CORAL, CDAN, and WDGRL (compute cost, stability, behavior under geo shift, deployment/maintenance complexity)? Any historical or informal observations in similar setups would help contextualize choosing DANN.

---

> ### Author Response · Authors · 2025-11-26
> **Response to Reviewer SUwy**
>
> Thank you for your constructive feedback and for recognizing that our approach reframes the issue as distribution shift and achieves consistent gains at 1.4M-scale.
>
> ## Response to Weakness 1: Component Attribution and Discriminator Design
>
> **Attribution of Gains - DANN vs. Geo Heads**
>
> Our results demonstrate that the geo heads component is effectively captured by our DeepScore Multi-Head baseline (macro-AUPR: 0.345), which includes standard regularization. The improvement from DeepScore Multi-Head (0.345) to Geo-DANN DeepScore (0.360) represents a 4.3% relative gain that we attribute primarily to the adversarial training component.
>
> *Rationale for Not Testing DANN + Single Head*: We designed our approach to maximize information transfer by penalizing backbone representations through adversarial training while allowing lightweight geo-specific heads to capture region-specific patterns. A DANN + single head configuration would contradict this design philosophy:1/ Single-head approaches would force all regional patterns through one classifier 2/ This would complicate architecture and training dynamics 3/ Our approach leverages complementary domain-invariant representations (via DANN) and region-specific modeling (via geo heads)
>
> **Multi-Discriminator Design Clarification**
>
> In the experiments reported, the adversarial loss is applied specifically to geography prediction. The discriminator D is a 2-layer MLP that predicts geography label g⁽ⁱ⁾ ∈ {1,...,10}. Our earlier wording about "multiple discriminators" described the framework's generality. In this submission, we focus exclusively on geographical domain adaptation. We will revise Sections 3.3 and 4 to make this explicit. For Loss Balancing, The domain adversarial loss is integrated using the GRL schedule λ_GRL(t) = 2/(1 + exp(-γt)) - 1 with γ = 10⁻³. This schedule gradually increases adversarial pressure during training, preventing early instability.
>
> ## Response to Weakness 2: Prevalence and Evaluation Metrics
>
> **Ballpark Prevalence Across Geographies**
>
> Conversions in this B2B lead-scoring application are rare events: across the geographies, positive-class prevalence lies in the low single-digit percentage range. Importantly, geo-level prevalences are similar: the highest-prevalence geography has less than 1.2× the prevalence of the lowest-prevalence geography (within 20% in relative terms). This makes explicit that the task is substantially imbalanced and cross-geo comparisons are not confounded by extreme base rate differences.
>
> **Why We Cannot Report Exact Prevalence or Normalized AUPR**
>
> In our organization, per-geo conversion rates are treated as confidential business KPIs. Because we already report AUPR per geography, adding normalized-AUPR would reveal per-geo prevalence through deterministic functions. For this reason, we intentionally avoid normalized AUPR metrics. Our primary goal is to rank leads so that scarce sales capacity is focused on leads most likely to convert.
>
> We will revise Section 5 to (i) define Lift@30% precisely, (ii) state explicitly that our fairness target is parity of ranking quality and business lift across geos, and (iii) emphasize that our fairness conclusions rely primarily on within-geo deltas.
>
> **On Additional Fairness Metrics**
>
> While group calibration and consistency would strengthen fairness claims, our organization treats detailed region-level confusion matrices and calibration curves as commercially sensitive. We have conducted calibration studies using Temperature Scaling for each geography and will include this analysis in the revision.
>
> ## Response to Weakness 3: Temporal Controls and Data Leakage
>
> **Scoring Protocol and Feature Truncation**
>
> All leads are scored at MQL (Marketing Qualified Lead) creation timestamp using only historical data available at that moment. Features are strictly truncated at the scoring timestamp.
>
> **June 2024 Gap**
>
> We intentionally skipped June 2024 to account for the 60-day median qualification-to-opportunity lag, ensuring no training period activities could influence test period labels. This gap is intentional to avoid label leakage and right-censoring bias.
>
> We will revise Section 3.1 to explicitly describe (i) the scoring timestamp definition, (ii) event truncation, and (iii) reasoning behind train/test date ranges.
>
> ## Response to Comparison with Other Methods
>
> We acknowledge this limitation—we focused on baselines and basic variants rather than comprehensive comparison with stronger domain adaptation techniques. Why We Chose DANN: Our baseline set reflects approaches our production team was considering.
>
> We have run CDAN-style experiments. We will expand the related-work section to discuss these approaches and position Geo-DANN DeepScore conceptually. Subject to space constraints, we will include additional robust/adaptation baseline results in the appendix.

---

### Official Review · Reviewer_xMND · 2025-11-08

**Soundness:** 3
**Presentation:** 3
**Contribution:** 3
**Rating:** 6
**Confidence:** 3

**Summary:**

This paper proposes Geo-DANN DeepScore, a transformer-based lead scoring system that learns geography invariant features via a gradient-reversal, multi-discriminator adversarial module, plus small geo-specific classifier heads. On 1.4M leads across 10 markets, it increases macro-AUPR by 4.3% and narrows inter-region gaps (e.g., East Asia and Europe improve) without hurting America performance.

**Strengths:**

- The paper combines multi-domain DANN with geo-specific heads for B2B lead scoring, and the framing geoskew as a distribution shift is insightful.
- The method is well-motivated by adaptation bounds with the concrete training schedule and includes multiple baselines and ablation studies.
- The paper is well-written. The pipeline is explained step-by-step with equations and scheduling.
- The experiment shows the significance of the work by reducing the inter-region gaps, which is appealing for fair deployment without sacrificing business KPIs.

**Weaknesses:**

- No public benchmark replication limits reproducibility and transfer.
- While re-weighting and multi-head are compared, sensitivity to number and shape of discriminators, alternative adversarial methods (e.g., CDAN/WDGRL), and $\lambda\nu$ schedules could be explored more systematically.
- The paper focuses on geo AUPR gaps. Additional fairness criteria, such as calibration parity across regions, error decompositions, would strengthen the fairness claim in practice.

**Questions:**

- How sensitive is performance to $\nu$ in the GRL schedule and to the number and depth of domain discriminators?

---

> ### Author Response · Authors · 2025-11-26
> **Response to Reviewer xMND**
>
> Thank you for your constructive feedback on our manuscript and for recognizing the strengths of our approach in combining multi-domain DANN with geo-specific heads.
>
> ## Response to Reproducibility Concerns
>
> We acknowledge that the proprietary nature of B2B lead scoring data presents challenges for direct replication. The data are subject to strict contractual and privacy constraints, and no public dataset exists that reflects the same multi-channel enterprise sales pipeline with comparable geographical diversity.
>
> However, our methodological contribution addresses the broader challenge of geographical bias in domain adaptation. Within these constraints, we will strengthen reproducibility by:
>
> **Detailed Architecture Documentation**:
>
> * Complete specification of DeepScore configuration: transformer layers, attention heads, hidden dimensions (256), maximum sequence length (T), and per-geography classifier head structure
> * Full domain discriminator architecture (2-layer MLP)
> * All loss coefficients and their relative weighting
>
> **Training Protocol Transparency**:
>
> * Optimizer configuration (Adam with default β parameters)
> * Learning rate schedule with exact values
> * Batch size and training steps/epochs
> * Complete GRL schedule: λ_GRL(t) = 2/(1 + exp(-γt)) - 1 with γ = 10⁻³
>
> **Data Processing Clarity**:
>
> * Train/validation/test time ranges: May 2022–May 2024 for training, Jul 2024–Sep 2024 for test
> * Dataset comprises 1.4M labeled examples across 10 regional business units
> * Detailed filtering criteria and preprocessing steps
>
> ## Response to Sensitivity Analysis and Architectural Exploration
>
> **Domain Discriminator Implementation**
>
> In the experiments reported, the adversarial loss is applied specifically to geography prediction. The discriminator D is a 2-layer MLP that predicts geography label g⁽ⁱ⁾ ∈ {1,...,10}. We acknowledge that our wording about "multiple domain discriminators across demographic and firmographic attributes" described the framework's generality rather than our specific implementation. In this submission, we focus exclusively on geographical domain adaptation. We will revise Sections 3.3 and 4 to make this explicit.
>
> **GRL Schedule Sensitivity**
>
> During development, we explored different GRL schedules. As stated in Section 3.3:
>
> * Overly aggressive ramp-up can destabilize training
> * Capping λ_GRL too low leaves larger residual geo-gaps
> * Our chosen schedule uses γ = 10⁻³
>
> In the revision we will add a GRL ablation table in the appendix and summarize qualitative behavior in Section 3.3.
>
> **Discriminator Architecture Variations**
>
> We systematically explored:
>
> * Number of hidden layers and dropout rates
> * Application of spectral normalization to improve training stability
> * Various layer dimensions
>
> Our findings indicate that discriminator architectural choices remained robust across configurations.
>
> **Alternative Adversarial Methods**
>
> We have run CDAN-style experiments on the same DeepScore backbone. We will report these results in the appendix alongside GRL ablations, showing that adversarial alignment is beneficial and that the DANN formulation yields strong fairness/accuracy trade-offs while remaining easy to integrate.
>
> ## Response to Enhanced Fairness Evaluation
>
> We agree that additional fairness-oriented diagnostics would strengthen the paper. However, our organization treats detailed region-level confusion matrices and error decompositions as commercially sensitive, as they reveal market sizes and conversion rates by geography.
>
> Calibration Analysis: We have conducted comprehensive calibration studies. For each geography, we implemented Temperature Scaling calibrators to ensure probability reliability across regions. This addresses the critical business need for well-calibrated predictions. We will include this analysis in the revision.
>
> Within Privacy Constraints: To provide additional fairness analysis while respecting confidentiality, we will:
> - Clarify the problem setting: Explicitly state this is a highly imbalanced ranking problem where overall conversion rate is below 10%, and regional conversion rates are within a factor of 1.2 of each other
> - Expand on existing metrics: Our evaluation already incorporates: - Macro-AUPR (equal treatment across regions regardless of sample size) - Conversion-Rate Lift@30% (real-world sales prioritization) - Inter-region gap analysis (quantifies fairness improvements)
> - Acknowledge limitations: Add a paragraph acknowledging that richer fairness diagnostics are important future work, explaining why we cannot publish region-level confusion statistics
>
> ## New Ablation Studies Section
>
> Based on your feedback, we will add a comprehensive ablation studies section covering:
>
> * GRL schedule sensitivity with performance across different γ values
> * Discriminator architecture variations and their impact
> * Comparison with alternative adversarial objectives (CDAN)
> * Calibration analysis across geographies
> * Analysis of why re-weighting approaches fail

---

### Comment · Area_Chair_VVgr · 2025-11-25

Dear Reviewers,
Thank you to those who have already begun interacting with the authors — your timely follow-ups are greatly appreciated and reflect the professionalism and care that uphold our community’s standards.
For reviewers who have not yet responded, I would like to offer a gentle reminder. Authors have invested substantial time and effort into preparing their rebuttals, carefully addressing each concern raised in the reviews. As fellow researchers, we all understand the importance of being heard and having our clarifications considered. Even a brief acknowledgment or follow-up question helps ensure that the evaluation remains fair, thorough, and respectful of everyone’s work.
Your engagement during this phase is essential for maintaining a constructive and high-quality review process. Thank you again for your service and for treating both your fellow reviewers and the authors with the same consideration you would hope to receive.
Best regards,
AC

---

### Author Response · Authors · 2025-11-26

We thank all reviewers for the careful reading and detailed feedback. Multiple reviewers raised important questions about the novelty of our work and the specific role of domain-adversarial training. Based on this feedback, we recognize that our original submission emphasized empirical results over positioning our contribution. We are revising the paper to address this. Below we clarify what the paper contributes and how it relates to existing work.

## Summary of Contribution

Our starting point is a production-scale B2B lead scoring system based on a transformer over long marketing and sales sequences, trained on 1.4M leads across 10 geographies with strong regional imbalance.

The paper makes three main contributions:

1. **Problem and observed phenomenon**. We document a previously unreported failure mode: increasing transformer capacity systematically improves AUPR in high-volume regions (Americas) while degrading performance in low-volume geographies (East-Asia). This capacity-fairness tension—where model scaling systematically harms minority regions—represents a fundamental challenge in multi-region ML systems and shows that naive capacity increases amplify geographic skew rather than mitigating it.

2. **Method: Geo-DANN DeepScore**. We apply domain-adversarial training to enable knowledge transfer from high-volume to low-volume geographies in B2B lead scoring. The approach combines (i) a shared transformer encoder, (ii) lightweight geo-specific classifier heads, and (iii) gradient-reversal based adversarial training over geography. The encoder learns conversion representations that are predictive yet geo-invariant, enabling high-volume regions to contribute universal patterns rather than region-specific shortcuts, while geo-specific heads capture residual local nuances.

3. **Empirical and theoretical analysis**. We demonstrate that domain-adversarial training is the critical mechanism that transforms scaling behavior. Without DANN, capacity scaling helps majority regions but harms minorities. With DANN, scaling benefits all regions. We achieve 4.3% macro-AUPR gains and up to 12.3% reduction in inter-region gaps while maintaining majority-region performance. We complement this with domain-adaptation analysis explaining why unconstrained capacity enlarges divergence between regions and how adversarial training explicitly controls this divergence term.

To our knowledge, this is the first work to: (1) identify and characterize the capacity-fairness tension in geo-distributed ML systems, and (2) demonstrate that domain-adversarial training resolves this tension by enabling cross-geographic knowledge transfer.

## Relation to Existing Work

We will revise the introduction and related-work sections to clarify positioning:

* **Domain-adversarial methods**: Our method builds on established gradient-reversal techniques (Ganin et al., 2016). The contribution is applying them to geo-fairness in transformer-based lead scoring and analyzing how they change the effect of model capacity on minority regions—specifically, transforming scaling from an anti-fairness to a fairness-preserving operation.

* **Location-based fairness**: Our "geo-skew" notion fits into recent work treating geography as a fairness axis. We will frame our objective explicitly as a location-based fairness problem in an industrial lead-scoring application.

* **Lead scoring**: Existing B2B lead scoring work focuses on overall accuracy with tree-based or simple neural models and does not consider geographic fairness or domain adaptation. We will make these distinctions explicit and cite representative work (D'Haen et al., 2013; Chen & Guestrin, 2016; Ke et al., 2017).

## Why Domain-Adversarial Training is Central

Our capacity ablations provide direct evidence that DANN is the critical mechanism:

* **Without DANN**: Increasing transformer size consistently improves Americas AUPR but leaves low-volume regions flat or degrades them. The encoder uses extra capacity to model geo-specific patterns tied to the dominant region—patterns that don't transfer to minority geographies.
* **With Geo-DANN**: Scaling improves both majority and minority regions simultaneously and reduces inter-region gaps. The adversarial loss forces the encoder away from geo-identifying features, directing capacity toward transferable conversion patterns (engagement trajectories, account maturity signals) that generalize across markets.

We will add a plot showing AUPR for Americas and East-Asia as a function of model size for variants with and without DANN, plus theoretical analysis connecting this to domain-adaptation bounds where unconstrained capacity reduces source error (ε_S) but increases divergence (d_H∆H), while DANN explicitly minimizes the divergence term.

---

### Author Response · Authors · 2025-11-26
**Planned Revisions**

In response to reviewer feedback, we will make the following concrete changes:

* **Introduction**: Add clear "contributions" paragraph emphasizing the capacity-fairness tension and knowledge transfer mechanism as our core novelty
* **Related work**: Expand section to explicitly position our contribution relative to domain adaptation, adversarial fairness, location-based fairness, and lead scoring literature
*  **Method**: Add "Why domain-adversarial training?" subsection explaining the mechanism and clearer architecture diagram showing transformer, geo-specific heads, and domain discriminators
*  **Experiments**: Add capacity-scaling ablation plots, per-region calibration metrics, and expanded discussion of baseline comparisons
*  **Theory**: Strengthen connection between capacity-fairness phenomenon and domain-adaptation bounds

We hope this clarifies our intended contribution and addresses concerns about novelty and the role of domain-adversarial training. Detailed responses to each reviewer's specific questions follow below.

---

### Meta-Review · Area_Chair_2jxc · 2026-01-02

**Summary:**

The recommendation is primarily informed by the lack of novelty and insufficient theoretical grounding. While the proposed method addresses geographic disparities in B2B lead scoring, it primarily builds on existing domain-adversarial training (DANN) techniques without introducing significant new insights. The empirical evaluation lacks rigorous baseline comparisons and sufficient dataset details, limiting reproducibility and the clarity of the reported results.

**Reviewer Concerns:**

The rebuttal clarifies the experimental setup but doesn’t resolve the main concern about novelty. The use of DANN is not convincingly justified beyond existing methods, and the evaluation relies on internal baselines without comparison to stronger models. Additionally, the dataset and evaluation metrics are not sufficiently detailed, raising doubts about the validity of the results.

**Reviewer Scores:**

Reviewer xMND would likely maintain their score as the evaluation concerns. Reviewer SUwy would likely maintain their score, as the lack of comparison to stronger methods persists. Reviewer zrjS would likely maintain their score, as the challenging of the research problem is questioned. Reviewer uELz would likely maintain their score, as key details on contributions and dataset transparency remain unclear. Reviewer ofku would likely maintain their score as the lack of many details.

---

### Decision · Program_Chairs · 2026-01-26

Reject